# Controlled Model Debiasing through Minimal and Interpretable Updates

**Federico Di Gennaro**[1,2]                                                    *federico.digennaro@epfl.ch*
*EPFL, Lausanne, Switzerland*

**Thibault Laugel**[1]                                                          *thibault.laugel@axa.com*
*AXA, Paris, France*
*TRAIL, LIP6, Sorbonne Université, Paris, France*

**Vincent Grari**
*AXA, Paris, France*
*TRAIL, LIP6, Sorbonne Université, Paris, France*

**Marcin Detyniecki**
*AXA, Paris, France*
*TRAIL, LIP6, Sorbonne Université, Paris, France*
*Polish Academy of Science, IBS PAN, Warsaw, Poland*

**Reviewed on OpenReview:** *https://openreview.net/forum?id=B9fdU4qjpD*

## Abstract

Traditional approaches to learning fair machine learning models often require rebuilding models from scratch, typically without considering potentially existing models. In a context where models need to be retrained frequently, this can lead to inconsistent model updates, as well as redundant and costly validation testing. To address this limitation, we introduce the notion of *controlled model debiasing*, a novel supervised learning task relying on two desiderata: that the differences between the new fair model and the existing one should be (i) minimal and (ii) interpretable. After providing theoretical guarantees to this new problem, we introduce a novel algorithm for algorithmic fairness, COMMOD, that is both model-agnostic and does not require the sensitive attribute at test time. In addition, our algorithm is explicitly designed to enforce minimal and interpretable changes between biased and debiased predictions in a binary classification task—a property that, while highly desirable in high-stakes applications, is rarely prioritized as an explicit objective in fairness literature. Our approach combines a concept-based architecture and adversarial learning and we demonstrate through empirical results that it achieves comparable performance to state-of-the-art debiasing methods while performing minimal and interpretable prediction changes. Code to reproduce the experiments is available on the following repository: https://github.com/axa-rev-research/controlled-model-debiasing

## 1 Introduction

The increasing adoption of machine learning models in high-stakes domains—such as criminal justice (Kleinberg et al., 2016) and credit lending (Bruckner, 2018)—has raised significant concerns about the potential biases that these models may reproduce and amplify, particularly against historically marginalized groups. Recent public discourse, along with regulatory developments such as the European AI Act (2024/1689), has further underscored the need for adapting AI systems to ensure fairness and trustworthiness (Bringas Colmenarejo et al., 2022). Consequently, many of the machine learning models deployed by organizations are,

---
[1]Equal contribution.
[2]Work carried out during an internship at AXA.

or may soon be, subject to these emerging regulatory requirements. Yet, such organizations frequently invest significant resources (e.g. time and money) in validating their models with the assistance of domain experts before deploying them at scale (e.g. Mata et al. (2021) for dam safety monitoring and Tsopra et al. (2021) for precision medicine), to ensure their performance and trustworthiness. Hence, as new regulatory constraints emerge, the ability to make these models comply with new constraints while minimizing the need for revalidation has thus become critically important.

The field of algorithmic fairness in a classification framework has experienced rapid growth in recent years, with numerous bias mitigation strategies proposed (Romei & Ruggieri, 2014; Mehrabi et al., 2021). These approaches can be broadly categorized into three types: *pre-processing* (e.g.,Belrose et al. (2024)), *in-processing* (e.g.,Zhang et al. (2018)), and *post-processing* (e.g., Kamiran et al. (2010)), based on the stage of the machine learning pipeline at which fairness is enforced. While the two former categories do not account at all for any pre-existing biased model being available for the task, post-processing approaches aim to impose fairness by directly modifying the predictions of a biased classifier: this naturally imposes some kind of consistency between the new, fair model and the old, biased one. However, these methods are generally deemed to achieve lower performance, and the changes performed are generally not traceable.

Motivated by these considerations, in this paper we consider a new paradigm, proposing to frame algorithmic fairness as a model update task. The goal is thus to enforce fairness through small and understandable updates to a pretrained model, presumably biased, in order to facilitate model monitoring. For this purpose, we introduce the notion of *Controlled Model Debiasing*, based on two intuitive desiderata: **(i)** changes between the new model and the existing one should be **minimal**, and **(ii)** these changes should be **understandable**. We stress the fact that requiring minimal updates ensures that we change as few existing *decisions* as possible, which preserves model stability, reduces the operational burden of re-validation, and maintains user trust in settings where consistency is critical. At the same time, enforcing interpretable updates means each model change can be clearly explained to domain experts and stakeholders—facilitating faster audit cycles, clearer accountability, and easier troubleshooting when fairness constraints interact with real-world business rules.

Our formulation combines assumptions from both *post-processing* and *in-processing* approaches, proposing to learn a new fair model by leveraging a previously trained biased one. After providing theoretical guarantees on the solution and feasibility of this new proposed problem, we introduce COMMOD (COncept-based Minimal MOdel Debiasing), our method to address it. Leveraging the fields of concept-based interpretability (Alvarez Melis & Jaakkola, 2018; Koh et al., 2020b) and fair adversarial learning (Zhang et al., 2018; Grari et al., 2021), COMMOD is a model-agnostic, interpretable debiasing method that aims to improve fairness in an interpretable way, while minimizing prediction changes relative to the original model. Through experiments, we demonstrate that our method achieves comparable fairness and accuracy performance to existing algorithmic fairness approaches, while requiring fewer prediction changes. Additionally, we show that COMMOD enables more meaningful and easier-to-understand prediction changes, enhancing its utility in practice. To summarize, our contributions are as follows:

- We introduce a new notion of Controlled Model Debiasing, which aims to account for a previously trained model by performing minimal and interpretable updates to mitigate bias (Section 3). To the best of our knowledge, we are the first to address interpretability directly in the updating process.

- We provide two theoretical guarantees for the proposed problem: the formulation of the Bayes-Optimal Classifier in a fairness-aware setting and the feasibility of the problem in the general setting (Section 4).

- We introduce a new method, COMMOD, to address the new optimization problem in a model-agnostic, fairness-unaware setting (Section 5). We validate its performance through experiments on classical fairness datasets, showcasing its debiasing efficacy and ability to perform fewer and more interpretable changes (Section 6).

# 2 Background and notation

Let $\mathcal{X} \subseteq \mathbb{R}^d$ be an input space consisting of $d$ features and $\mathcal{Y}$ an output space. In a traditional algorithmic fairness framework in supervised learning, we want an algorithm that outputs $\hat{Y} \in \mathcal{Y}$ such that $\hat{Y}$ is unbiased from a sensitive variable $S$, only available at training time. For the sake of simplicity, we focus in this paper on a binary classification problem where $\mathcal{Y} = \{0, 1\}$ and for which also the sensitive attribute $S$ takes values on a binary sensitive space $\mathcal{S} = \{0, 1\}$.

## 2.1 Group fairness

Over the years, several notions of fairness have been proposed in the literature to define whether or not a variable $\hat{Y}$ is unbiased from a sensitive variable $S$ (Dwork et al., 2012; Hardt et al., 2016; Jiang et al., 2020). In this paper, we focus on two of the most used ones: *Demographic Parity* (Calders et al., 2009) and *Equalizing Odds* (Hardt et al., 2016).

### 2.1.1 Demographic Parity

A classifier satisfies *Demographic Parity* (DP) if the prediction $\hat{Y}$ is independent from sensitive attribute $S$. In the case of a binary classification, this is equivalent to $\mathbb{P}(\hat{Y} = 1 \mid S = 0) = \mathbb{P}(\hat{Y} = 1 \mid S = 1)$. Hence, Demographic Parity can be measured using the *P-rule*:

$$P\text{-}Rule = \min\left(\frac{\mathbb{P}(\hat{Y} = 1 \mid S = 1)}{\mathbb{P}(\hat{Y} = 1 \mid S = 0)}, \frac{\mathbb{P}(\hat{Y} = 1 \mid S = 0)}{\mathbb{P}(\hat{Y} = 1 \mid S = 1)}\right).$$

### 2.1.2 Equalizing Odds

A classifier satisfies *Equalizing Odds* (EO) if the prediction $\hat{Y}$ is conditionally independent of the sensitive attribute $S$ given the actual outcome $Y$. In the case of a binary classification problem, this is equivalent to $\mathbb{P}(\hat{Y} = 1 \mid Y = y, S = 0) = \mathbb{P}(\hat{Y} = 1 \mid Y = y, S = 1)$, for all $y \in \{0, 1\}$. Hence, Equalizing Odds can be measured using the *Disparate Mistreatment* (DM) (Zafar et al., 2017):

$$\Delta_{TPR} = |TPR_{S=1} - TPR_{S=0}|, \quad \Delta_{FPR} = |FPR_{S=1} - FPR_{S=0}|,$$

where $TPR_{S=s} = \mathbb{P}(\hat{Y} = 1 | Y = 1, S = s)$, $FPR_{S=s} = \mathbb{P}(\hat{Y} = 1 | Y = 0, S = s)$, and $s \in \{0, 1\}$. In the rest of the paper, we use $DM = \Delta_{TPR} + \Delta_{FPR}$.

## 2.2 Mitigation strategies

Bias mitigation algorithms are generally grouped into three different categories, based on when the debiasing process is carried on in the machine learning pipeline (Romei & Ruggieri, 2014). In particular, *pre-processing* methods (e.g. Zemel et al. (2013)) modify the training data such that they are unbiased with respect to $S$. *In-processing* methods (e.g. Zhang et al. (2018)) aim to train a new, fair, classifier by integrating some fairness constraints in the optimization objective. Finally, *post-processing* methods aim to achieve fairness by adjusting the predictions $\hat{Y}^f$ (or the predicted probabilities) of an already existing model $f \colon \mathcal{X} \to \mathcal{Y}$. As such, a notion of predictive similarity between these two sets of predictions often being optimized, either directly (Jiang et al., 2020; Nguyen et al., 2021; Alghamdi et al., 2022) or indirectly through heuristics (Kamiran et al., 2018). However, rather than an assumed end-goal, this objective generally remains a proxy for the true goal of optimizing accuracy, as the true labels $Y$ are generally assumed to be unavailable at inference time in the post-processing setting. Furthermore, these approaches generally suffer from several limitations, such as not being model-agnostic (Calders & Verwer, 2010; Kamiran et al., 2010; Du et al., 2021), or requiring access to the sensitive attribute at test time (Hardt et al., 2016; Pleiss et al., 2017), which greatly hurt their practical use. On this topic, we provide a review of existing post-processing approaches and their limitations in Table 2 in Appendix.

# 3 Controlled model debiasing

Let us now consider a context where a model $f : \mathcal{X} \to [0, 1]$ that outputs each input $x \in \mathcal{X}$ to a predicted probability $\mathbb{P}(Y = 1 | X = x)$. Suppose now this model needs to be updated into a fairer model $g$ (with respect to a specific fairness definition discussed in Section 2.1) while trying to maintain the accuracy of $f$. As discussed in Section 1, blindly training $g$ can be harmful for several reasons. First, Krco et al. (2023) have shown that some bias mitigation approaches performed needlessly large numbers of prediction changes for similar levels of accuracy and fairness. Yet, a large number of inconsistent decisions may negatively impact users' trust, as shown by Burgeno & Joslyn (2020) in the context of weather forecast. On top of this, beyond statistical testing, models in production may undergo extensive testing by domain experts (Tsopra et al., 2021). In this context, being able to understand the changes from one model to the other would be crucial to not being forced to undergo all the testing again, in addition to increasing expert trust. Driven by the above-mentioned considerations, we introduce the notion of *Controlled Model Update*, which is based on the following notions of *Interpretable* and *Minimal* updates.

## 3.1 Interpretable updates

To facilitate the adoption of a new model $g$, we propose to generate explanations that describe how the debiasing process modified the old model $f$. Contrary to traditional XAI methods (see Guidotti et al. (2018) for a survey), rather than explaining the decisions of the model $g$, our aim here is to generate insights about the differences between $f$ and $g$. An intuitive solution to this problem would be to first train a fair model and then generating explanations for these differences in a post-hoc manner, akin to Renard et al. (2024). However, post hoc interpretability methods, in general, are often criticized for their lack of connection with ground-truth data (Rudin, 2019; Laugel et al., 2019). Therefore, in order to ensure better trust in the new model, we pursue the direction of self-explainable models (Alvarez Melis & Jaakkola, 2018), generating global explanations for the differences between $g$ and $f$ while learning the predictive task.

## 3.2 Minimal updates

The constraint of ensuring that the new model remains as similar as possible to the initial one can be directly added to the training objective. In particular, to the usual optimization problem of fairness, we added a constraints that control the probability for a prediction $\hat{Y}^f$ to change label after the debiasing process. We emphasize once again that making minimal changes to a biased model is beneficial in cases where the model has already been validated based on user needs. Significant updates to make the model fairer might risk no longer satisfying those needs, requiring once again a costly validation. We then frame our problem in a traditional algorithmic-fairness way, in which the self-explainable model $g$ is required to be **accurate** and **fair**. On top of these constraints, $g$ is required to be **similar** to $f$ according to a distance function $\text{dist}_\phi : [0, 1] \times [0, 1] \to \mathbb{R}_{\geq 0}$, where $\text{dist}_\phi(f, g) = \mathbb{E}_{X \sim \mathcal{D}}[\phi(f(X), g(X))]$. The combination of these desiderata allows us to understand what changes are required from a biased model to become fairer focusing on minimal adjustments.

Let $\text{fair}(\cdot)$ be the fairness criteria (e.g. *Demographic Parity* or *Equalizing Odds*). Further, let us define the predictions (after threshold) of the biased model $f$ and edited model $g$ as $\hat{Y}^f = \mathbf{1}_{\text{f(X)}>0.5}$ and $\hat{Y} = \mathbf{1}_{\text{g(X)}>0.5}$ respectively. We can then formalize our optimization problem through the following *Controlled Model Update* objective:

$$\begin{aligned} \min \quad & Acc(\hat{Y}, Y) \\ \text{s.t.} \quad & \text{fair}(\hat{Y}) \geq \tau \\ & \text{dist}_\phi(f, g) \leq p \end{aligned} \quad (1)$$

where $\tau \in \mathbb{R}_+$ and $p \in \mathbb{R}$. One could also notice that, by fixing $\phi(f, g) = |\mathbf{1}_{(f>0.5)} - \mathbf{1}_{(g>0.5)}|$, the distance becomes $\text{dist}_\phi(f, g) = \mathbb{P}(\hat{Y}^f \neq \hat{Y})$. Through the paper, this is the distance we will refers to in our experiments, but a brief discussion on other possible distances (i.e. different $\phi$) is reported in Section 5.1 and Appendix D.4. Observe that setting $p = 0$, i.e. forbidding any prediction change to happen, imposes $g = f$. On the contrary, the more $p$ increases and the more $g$ is free to differ from $f$ and, ideally, to reach higher

fairness scores. When $p = 1$, Eq. 1 becomes a classical algorithmic fairness learning problem. Notice also that the interpretability is all carried by the self-explainable model $g$ itself, and hence it will be not explicitly appear in the optimization problem of Equation 1. We will discuss our self-explainable model $g$ in Section 5.2.

# 4 Theoretical guarantees on the minimal update problem

In this section, we present two theoretical contributions to the problem formalized in Eq. 1, where we used as distance $\text{dist}_\phi(f, g) = \mathbb{P}(\hat{Y}^f \neq \hat{Y})$. First (Section 4.1), we show that it is possible to define the Bayes Optimal Classifier of the problem in the fairness-aware setting. Then, in Section 4.2, we assess the feasibility of the problem in general, giving guarantees on the trade-off between fairness and number of prediction changes. All proofs can be found in Appendix B.

## 4.1 Result 1: Bayes-optimal classifier in a fairness-aware setting

Let $(X, S, Y, \hat{Y}^f) \sim \mathcal{D}_{jnt}$ be a joint distribution on $\mathcal{X} \times \{0, 1\} \times \{0, 1\} \times \{0, 1\}$, where $X$ is the instance, $Y$ the target feature, $S$ the sensitive variable, and $\hat{Y}^f$ the output of the black-box classifier. Let $\mathcal{D}$ be the distribution over $(X, Y)$, $\bar{\mathcal{D}}$ a suitable distribution (DP or EO) over $(X, S)$ and $\mathcal{D}^*$ a distribution over $(X, \hat{Y}^f)$. We want to show that despite the new constraint on the number of changes, our optimization problem in Eq. 1 still has an analytical solution (i.e. the Bayes-optimal classifier (BOC)) in a fairness aware setting knowing the true distributions above mentioned. To do so, we leverage the work of (Menon & Williamson, 2018), who defined the BOC in the traditional fairness setting, relying on a reformulation of the optimization problem using the notion of cost-sensitive risk[1]. For what concerns the fairness constraint, they show (Lemma 3) that $P\text{-}Rule(g) \geq \tau$ is equivalent to $CS_{bal}(g, \bar{\mathcal{D}}, \bar{c}) \geq k$, where $CS_{bal}(g, \bar{\mathcal{D}}, \bar{c}) = (1 - \bar{c}) \cdot FNR(g; \bar{\mathcal{D}}) + \bar{c} \cdot FPR(g; \bar{\mathcal{D}})$ is the *balanced cost-sensitive risk* and $\bar{c} = 1/(1 + \tau)$.

Following the same idea, we first establish the equivalence between the constraint on the number of changes and a cost-sensitive risk:

**Lemma 4.1.** *Pick any random classifier $g$. Then, for any $p \in [0, 1]$,*

$$\mathbb{P}(\hat{Y}^f \neq \hat{Y}) \leq p \Leftrightarrow CS_{bal}(g; \mathcal{D}^*, c^*) \leq p,$$

*with $CS_{bal}(g; \mathcal{D}^*, c^*) = (1 - c^*) \cdot FNR(g; \mathcal{D}^*) + c^* \cdot FPR(g; \mathcal{D}^*)^2$ and $c^* = \mathbb{P}(\hat{Y} = 1)$.*

Finally, in a similar fashion of the main result in (Menon & Williamson, 2018) about the Bayes-Optimal Classifier, we can define for any cost $c$, $\bar{c}$, $c^* \in [0, 1]$, $\lambda_{fair}$, $\lambda_{ratio} \in \mathbb{R}$ the Lagrangian $R_{CS}(g; \mathcal{D}, \bar{\mathcal{D}}, \mathcal{D}^*)$ of the optimization problem in Eq. 1 as:

$$R_{CS}(g; \mathcal{D}, \bar{\mathcal{D}}, \mathcal{D}^*) = CS(g; \mathcal{D}, c) - \lambda_{fair} CS_{bal}(g; \bar{\mathcal{D}}, \bar{c}) \tag{2}$$
$$- \lambda_{ratio} CS_{bal}(g; \mathcal{D}^*, c^*).$$

**Proposition 4.2** (Fairness-aware Bayes-optimal classifier)**.** *The Bayes optimal classifier of $R_{CS}$ in a fairness aware setting and knowing the distributions $\mathcal{D}, \bar{\mathcal{D}}, \mathcal{D}^*$ is given by*

$$g_{opt}(x) = \underset{g \in [0,1]^{\mathcal{X}}}{\arg\min} R_{CS}(g; \mathcal{D}, \bar{\mathcal{D}}, \mathcal{D}^*) = \{H_\eta \circ s^*(x) \mid \alpha \in [0, 1]\}$$

*where*

$$\eta(x) = \mathbb{P}(Y = 1 \mid X = x), \ \bar{\eta}(x) = \mathbb{P}(S = 1 \mid X = x), \ \eta^*(x) = \mathbb{P}(\hat{Y}^f = 1 \mid X = x),$$
$$s^*(x) = \eta(x) - c - \lambda_{ratio}(\bar{\eta}(x) - \bar{c}) - \lambda_{fair}(\eta^*(x) - c^*),$$

*and $H_\eta(z) = \mathbf{1}_{\{z > 0\}} + \alpha \mathbf{1}_{\{z = 0\}}$ is the modified Heaviside (or step) function for a parameter $\eta \in [0, 1]$.*

---

[1]From Menon & Williamson (2018): The risk of a randomized classifier $g \colon \mathcal{X} \to (0, 1)$ computed on $Y$ is called *cost sensitive risk* for a cost parameter $c \in (0, 1)$ and $\pi = \mathbb{P}(Y = 1)$ if it can be written as $CS(g, \mathcal{D}, c) = \pi \cdot (1 - c) \cdot FNR(g; \mathcal{D}) + (1 - \pi) \cdot c \cdot FPR(g; \mathcal{D})$, where $FNR(g; \mathcal{D}) = \mathbb{E}_{X|Y=1}[1 - g(X)]$ and $FPR(g; \mathcal{D}) = \mathbb{E}_{X|Y=0}[g(X)]$.

[2]Notice that we changed the distribution w.r.t. FPR and FNR are computed. Hence, $FNR(g; \mathcal{D}^*) = \mathbb{P}(\hat{Y} = 0 | \hat{Y}^f = 1)$ and $FPR(g; \mathcal{D}^*) = \mathbb{P}(\hat{Y} = 1 | \hat{Y}^f = 0)$.

Despite its theoretical importance, the BOC presented in Proposition 3.2 remains impractical for computation, as the joint and marginal distributions of the random variables $X, S, Y, \hat{Y}^f$ are typically inaccessible in real-world applications. Furthermore, the above result only holds within a fairness-aware setting, a context that is uncommon in practical scenarios. For this reason, we introduce in Sec. 5 a novel algorithm that optimizes a relaxed form of the optimization problem in Eq. 1, which does not require knowledge of the sensitive attribute at test time. Finally, note that this result can be directly adapted to EO, since the additional constraint we bring does not depend on the fairness criteria, and Menon & Williamson (2018) also applies to EO.

## 4.2 Result 2: fairness level under a maximum change constraint

We now aim to theoretically understand what happens experimentally to the trade-off between fairness score and the similarity between $f$ and $g$ defined in Eq. 1. For this purpose, we study the impact that $K$ changes to the predictions of $f$ can have on *P-Rule* and on Disparate Mistreatment (DM).

Let us consider a set $\{(X_i, Y_i, S_i)\}_{i=1}^N$, with $S_i \in \{0, 1\}$ and $Y_i \in \{0, 1\}$. Then, we can define the following quantities

$$\gamma_{s1}(f) = \left|\{\, i : S_i = s, \ f(X_i) = 1\}\right|, \quad \gamma_{s0}(f) = \left|\{\, i : S_i = s, \ f(X_i) = 0\}\right|,$$

where $|\{set\}|$ stays for the size of such a set. Then, with $C = S_0/S_1$, the empirical P-Rule of $f$ is

$$\text{P-Rule}(f) = \min\left(C \cdot \frac{\gamma_{11}(f)}{\gamma_{10}(f)}, \ \frac{1}{C} \cdot \frac{\gamma_{10}(f)}{\gamma_{11}(f)}\right).$$

**Definition 4.3.** (Switching point) The switching point $K_s$ is the minimum number of changes required to get a P-Rule$(g) \approx 1$ starting from P-Rule$(f)$.

Informally, the switching point refers to the phenomenon in which, starting from one of the two functional forms in the minimum definition of the P-Rule, we switch to the reciprocal form definition. With this in mind, we can prove the following proposition:

**Proposition 4.4** (Extreme-flip optimality for Demographic Parity). *Given $K$ changes with $K < K_s$, the maximum P-Rule we can get by editing with $K$ flips the predictions of a model $f$ is achieved by either dedicating all the flips to one side of the division $\gamma_{11}(f)$ vs $\gamma_{10}(f)$.*

Note that, in minimal changes regimes of these paper, the it is reasonable to assume that $K < K_s$. We believe this result represents a significant milestone in the theoretical characterization of the effects of debiasing a model using DP as a fairness measure. Although this result provides valuable insights into the *dynamics* of the debiasing process, Proposition 4.4 does not account for the potential impact of these optimal changes on model accuracy. This highlights the possibility that, in the context of algorithmic fairness—where the goal is to balance fairness and accuracy—debiasing algorithms may prioritize alternative changes that minimize adverse effects on accuracy. On top of this, this theoretical result would be algorithmically applicable only in a fairness aware setting due to the fact that the knowledge of proportion $\gamma_{ij}, \ i, j \in \{0, 1\}$ is required.

**Extension of the result to Equalized Odds** . Further, this result could be also presented for EO and a proof follows more or less the same argument. In fact, we can define

$$N_{s,y} \ = \ \left|\{\, i : S_i = s, \ Y_i = y\}\right|, \qquad s, y \in \{0, 1\},$$

i.e. the number of instances in group $s$ with label $y$. The core insight is that, when you split $K$ flips between TPR-adjusting and FPR-adjusting actions, the resulting DM is a piecewise-linear function of the allocation. A convex (affine) function on a compact interval always achieves its minimum at one of the endpoints and hence the best you can do is to devote all $K$ flips to whichever single action gives the larger per-flip reduction in disparity. In particular, we can prove the following:

**Proposition 4.5** (Optimal allocation for minimizing Disparate Mistreatment). *Let $f$ be a fixed base classifier and define*

$$\gamma = \max_{s \in \{0,1\}} \frac{1}{N_{s,1}}, \qquad \delta = \max_{s \in \{0,1\}} \frac{1}{N_{s,0}}.$$

*For any integer $K \geq 1$, consider all post-processed classifiers obtained by flipping exactly $K$ labels of $f$. If $x \in [0, K]$ denotes the number of flips devoted to reducing the TPR-gap (and $K - x$ those for the FPR-gap), the minimum Disparate Mistreatment over $x \in \{1, \ldots, K\}$ is attained at an endpoint:*

$$x_{\mathrm{opt}} = \begin{cases} 0, & \delta > \gamma, \\ any\ x \in [0, K], & \delta = \gamma, \\ K, & \delta < \gamma. \end{cases}$$

This means that the best strategy is to devote all $K$ flips to whichever type of flip gives the larger per-flip reduction.

## 5  COMMOD: an algorithm for controlled model debiasing

As previously mentioned, while the BOC of Proposition 4.2 is theoretically relevant in the Fairness-Aware setting, it lacks practical utility. In order to solve Problem 1, we thus propose a new algorithm, called COM-MOD, circumventing these limitations and integrating the interpretability objective described in Section 3. COMMOD aims to edit the probabilities scores $f(X)$ of the biased model into scores $g(X)$, whose associated predictions $\hat{Y} = \mathbb{1}_{g(X) > 0.5}$ are more fair and whose predictive quality is measured through a loss function $\mathcal{L}_Y(g(X), Y)$ (e.g. the binary cross-entropy loss). The update is done through a multiplicative factor $r(X)$ as follows:

$$g(X) = \sigma(r(X) f_{\mathrm{logit}}(X)), \tag{3}$$

where $\sigma(\cdot)$ is the sigmoid function to ensure $g(X) \in [0, 1]$ and $f_{\mathrm{logit}}$ is the logit value of the biased model $f$. Besides being intuitive, this formulation enables a more direct modelling of the differences between $f$ and $g$, as $\hat{Y} \neq \hat{Y}^f$ if and only if $r(X) < 0$. We then propose to model the ratio $r_{w_g}$ as a Neural Network with parameters $w_g$ and taking as input $X$. In the subsections below, we describe how both desiderata of **similarity** and **interpretability** are integrated in COMMOD.

### 5.1  Penalization term for minimal updates

We propose to minimize model changes by adding a penalization term $\mathcal{L}_{\mathrm{ratio}}$. Numerous similarity measures between $f$ and $g$ can be considered, depending on the considered scenario: for instance, existing works in post-processing fairness generally focus on distances between distributions such as the Wasserstein distance (Jiang et al., 2020) or the KL-divergence (Ţifrea et al., 2023). Here, we propose to use either a mean squared norm function $\mathcal{L}_{\mathrm{ratio}}(r_{w_g}(X)) = \|r_{w_g}(X) - \mathbb{1}\|_M^2$ defined by a positive-definite inner product $M$. This term ensure that the score $f(X)$ is not modified unnecessarily, and thus that the calibration of $g$ remains consistent (cf. Figure 9 in App. D.4 for a related discussion). For the sake of simplicity, in the experiments of our paper we set $M = I$, recovering the classical Euclidean distance. More generally, by choosing $M = \mathrm{diag}(w_1, \ldots, w_d) \succ 0$, one recovers a *weighted* Euclidean norm $\|r_{w_g}(X) - \mathbb{1}\|_M^2 = \sum_{i=1}^d w_i (r_{w_g}(X)_i - 1)^2$. Similarly, setting $M = \Sigma^{-1}$, with $\Sigma$ the (positive-definite) feature covariance matrix, yields the *Mahalanobis* distance $(r_{w_g}(X) - \mathbb{1})^\top \Sigma^{-1} (r_{w_g}(X) - \mathbb{1})$. In likelihood-based settings one can even take $M$ to be the Fisher-information matrix, providing a local quadratic approximation to the KL divergence.

### 5.2  Concept-based Architecture for Interpretable Changes

In our approach, we prioritize interpretability by utilizing a self-explainable model. *Sparse Linear Regression* are generally considered to be interpretable. On top of that, inspired by the growing body of works on *concepts* (Koh et al., 2020a; Yuksekgonul et al., 2023; Fel et al., 2023; Zarlenga et al., 2023), we propose to learn $k$ concepts $C : \mathcal{X} \to \mathbb{R}$ in an unsupervised manner where every concept is a **sparse** linear combination of the input features. Consistently with previous works, we also posit that these concepts should be both **diverse** (Alvarez Melis & Jaakkola, 2018; Fel et al., 2023; Zarlenga et al., 2023) to be meaningful and the least redundant possible. Hence, the model $g$ is interpretable due to the structure of $r_{w_g}$ that is a neural network without activation functions. The final architecture $r_{w_g}$ consists of an initial (linear) bottleneck that maps the input features $X$ into $k$ concepts, followed by an output layer that linearly combines these concepts to produce

the multiplicative ratio needed for the update. By using linear combinations for both the features-to-concepts mapping and the concepts-to-output mapping, we can directly examine the learned weights to understand the direction (positive or negative) in which each concept—and, by extension, each feature—contributes to the value of the ratio. This level of interpretability was not achievable in previous models like TabCBM (Zarlenga et al., 2023). To ensure that the learned concepts are meaningful, we introduce penalization terms for **diversity** and **sparsity**, similar to those considered in the existing interpretability literature (Zarlenga et al., 2023; Yuksekgonul et al., 2023; Khurana & Galhotra, 2021). Given our linear architecture, these terms are defined as follows: for sparsity, we use a Lasso penalization term, and for diversity, we define the penalization as $\mathcal{L}_{\text{diversity}} = \sum_{i,j \leq k} \rho(W^i, W^j)$, where $\rho$ denotes the cosine distance and $W^i, W^j$ are the network's weights from input features to concepts $i$ and $j$ respectively. The combined penalization term is then given by $\mathcal{L}_{\text{concepts}} = \mathcal{L}_{\text{diversity}} + \mathcal{L}_{\text{sparsity}}$.

Beyond ensuring interpretability, using a linear transformation rather than a more complex architecture stems from our observations that it often lead to comparable results along our metrics of interest (see D.3 in the Appendix) - an observation also made by (Ţifrea et al., 2023).

### 5.3 Final model

To enforce fairness, we propose to leverage the technique proposed by Zhang et al. (2018), which relies on a dynamic reconstruction of the sensitive attribute from the predictions using an adversarial model $h_{w_h} : \mathcal{Y} \rightarrow \mathcal{S}$ to estimate and mitigate bias. The higher the loss $\mathcal{L}_S(h_{w_h}(r_{w_g}(X)f_{\text{logit}}(X)), S)$ of this adversary is, the more fair the predictions $g_{w_g}(X)$ are. Furthermore, this allows us to mitigate bias in a fairness-blind setting (i.e. without access to the sensitive attribute at test time), thus overcoming the limitation of most post-processing fairness methods (cf Table 2).

We then use our rescaling network $r_{w_g}$, implemented as a purely linear mapping without activation functions (cf. Section 5.2); on the other hand, both the adversary $h_{w_h}$ and the predicting network $g_{w_g}$ are standard fully connected networks with non-linear activations. Thus, the relaxed version of Equation 1 for DP can be written as:

$$
\begin{aligned}
\min_{w_g} \quad & \mathbb{E}[\mathcal{L}_Y(g_{w_g}(X), Y)] \quad [1] \\
\text{s.t} \quad & \mathbb{E}[\mathcal{L}_S(h_{w_h}(r_{w_g}(X)f_{\text{logit}}(X)), S)] \geq \epsilon' \quad [2] \\
\text{and} \quad & \mathbb{E}[\mathcal{L}_{\text{ratio}}(r_{w_g}(X))] \leq \eta' \quad [3]
\end{aligned}
\tag{4}
$$

with $\epsilon', \eta' > 0$. In practice, we relax Problem 4, integrating the interpretability constraints on the concepts:

$$
\begin{aligned}
\arg\min_{w_g}\max_{w_h} \ \frac{1}{N} \sum_{i=1}^{N} & \mathcal{L}_Y(g_{w_g}(x_i), y_i) \\
& - \lambda_{\text{fair}}\mathcal{L}_S(h_{w_h}(r_{w_g}(x_i)f_{\text{logit}}(x_i)), s_i) \\
& + \lambda_{\text{ratio}}\mathcal{L}_{\text{ratio}}(r_{w_g}(x_i)) \\
& + \lambda_{\text{concepts}}\mathcal{L}_{\text{concepts}}(r_{w_g}(x_i)),
\end{aligned}
\tag{5}
$$

with $\lambda_{\text{fair}}$, $\lambda_{\text{ratio}}$ and $\lambda_{\text{concepts}}$ three hyperparameters and with $\mathcal{L}_Y$ and $\mathcal{L}_S$ binary cross-entropy losses. Similarly to (Zhang et al., 2018) this loss function can easily be adapted to address EO by modifying the adversary $h_{w_h}$ to take as both the true labels $Y$ and the predictions $g_{w_g}(X)$.

## 6 Experiments

After describing our experimental setting, we propose in this section to empirically evaluate the two dimensions of the *controlled model update*: the ability to achieve accuracy and fairness with minimal (Section 6.2), and interpretable (Section 6.3) prediction changes.

| | Method | Q1 Fairness | | Q2 Fairness | | Q3 Fairness | Q4 Fairness | |
|---|---|---|---|---|---|---|---|---|
| | | Q3 Acc | Q4 Acc | Q2 Acc | Q3 Acc | Q2 Acc | Q1 Acc | Q2 Acc |
| Law School | Oracle (ROC) | 0.18 | 0.08 | 0.26 | - | - | 0.33 | - |
| | Oracle (LPP) | - | 0.08 | - | 0.12 | - | - | - |
| | LPP | 0.11 | 0.07 | 0.15 | - | 0.19 | 0.23 | 0.21 |
| | Zhang | 0.17 | 0.12 | 0.23 | 0.18 | 0.28 | 0.34 | 0.28 |
| | FRAPPE | - | 0.03 | 0.16 | - | 0.18 | **0.24** | 0.21 |
| | COMMOD | **0.07** | **0.01** | **0.10** | **0.08** | **0.13** | **0.24** | **0.18** |
| Compas | Oracle (ROC) | 0.02 | 0.03 | - | 0.05 | 0.06 | - | 0.09 |
| | Oracle (LPP) | **0.02** | **0.01** | - | **0.03** | **0.03** | - | **0.06** |
| | LPP | 0.08 | 0.04 | 0.14 | 0.12 | - | - | - |
| | Zhang | 0.09 | 0.07 | 0.16 | 0.12 | 0.14 | **0.20** | 0.16 |
| | FRAPPE | - | 0.18 | 0.24 | - | 0.21 | 0.28 | - |
| | COMMOD | 0.05 | **0.01** | **0.09** | 0.09 | 0.05 | **0.20** | **0.06** |

Table 1: Avg. percentage of prediction class changes for various quartiles of accuracy and fairness on the Law School and Compas dataset (DP). The missing values indicate that no model trained with the corresponding method ended up in this quartile definition.

## 6.1 Experimental protocol

### 6.1.1 Datasets

We experimentally validate two binary classification datasets, commonly used in the fairness literature (Hort et al., 2024): Law School (Wightman, 1998) and Compas (Angwin et al., 2016). The sensitive attribute for both datasets is *race*.

### 6.1.2 Competitors and baselines

To assess the efficacy of COMMOD, we first use an **in-processing debiasing method**, AdvDebias (Zhang et al., 2018), described in Section 2.1. We use the implementation provided in the AIF360 library (Bellamy et al., 2018). Additionally, as discussed in Section 6.2, **post-processing debiasing methods** also use a pretrained classifier as input. Yet, most methods are not directly comparable as they are not model-agnostic or require the sensitive attribute a test time. Two exceptions are LPP (Xian & Zhao, 2024) and FRAPPE (Ţifrea et al., 2023), which we use as competitors. Furthermore, although not directly comparable, we include the fairness-aware algorithms proposed by Xian & Zhao (2024) (that we name *Oracle (LPP)*) and ROC (Kamiran et al., 2012) (*Oracle (ROC)*) as baselines. As they directly use the sensitive attribute to make predictions, these two algorithms are expected to outperform all the others.

### 6.1.3 Set-up

After splitting each dataset in $\mathcal{D}_{\text{train}}$ (70%) and $\mathcal{D}_{\text{test}}$ (30%), we train our pretrained classifier $f$ to optimize solely accuracy. In these experiments, we use a Logistic Regression classifier from the scikit-learn library, but any other classifier could be used since COMMOD and the proposed competitors are model-agnostic. We then train COMMOD, the competitors and the baselines on $\mathcal{D}_{\text{train}}$, varying hyperparameter values for each method to achieve a range of fairness and accuracy scores over $\mathcal{D}_{\text{test}}$. For COMMOD, we set a fixed value for the number of concepts $k$: 2 for Law School and 5 for Compas. Further details on implementation are available in Section C of the Appendix.

## 6.2 Experiment 1: achieving fairness through minimal changes

In this first experiment, we aim to assess the ability of COMMOD to achieve competitive results in terms of fairness and accuracy while performing a lower number of changes. For this purpose, we measure the proportion $\mathcal{P}$ of prediction changes between the black-box model and the fairer model on the test set $\mathcal{D}_{\text{test}}$. Although COMMOD optimizes a continuous similarity metric over probabilities, for comparability with the

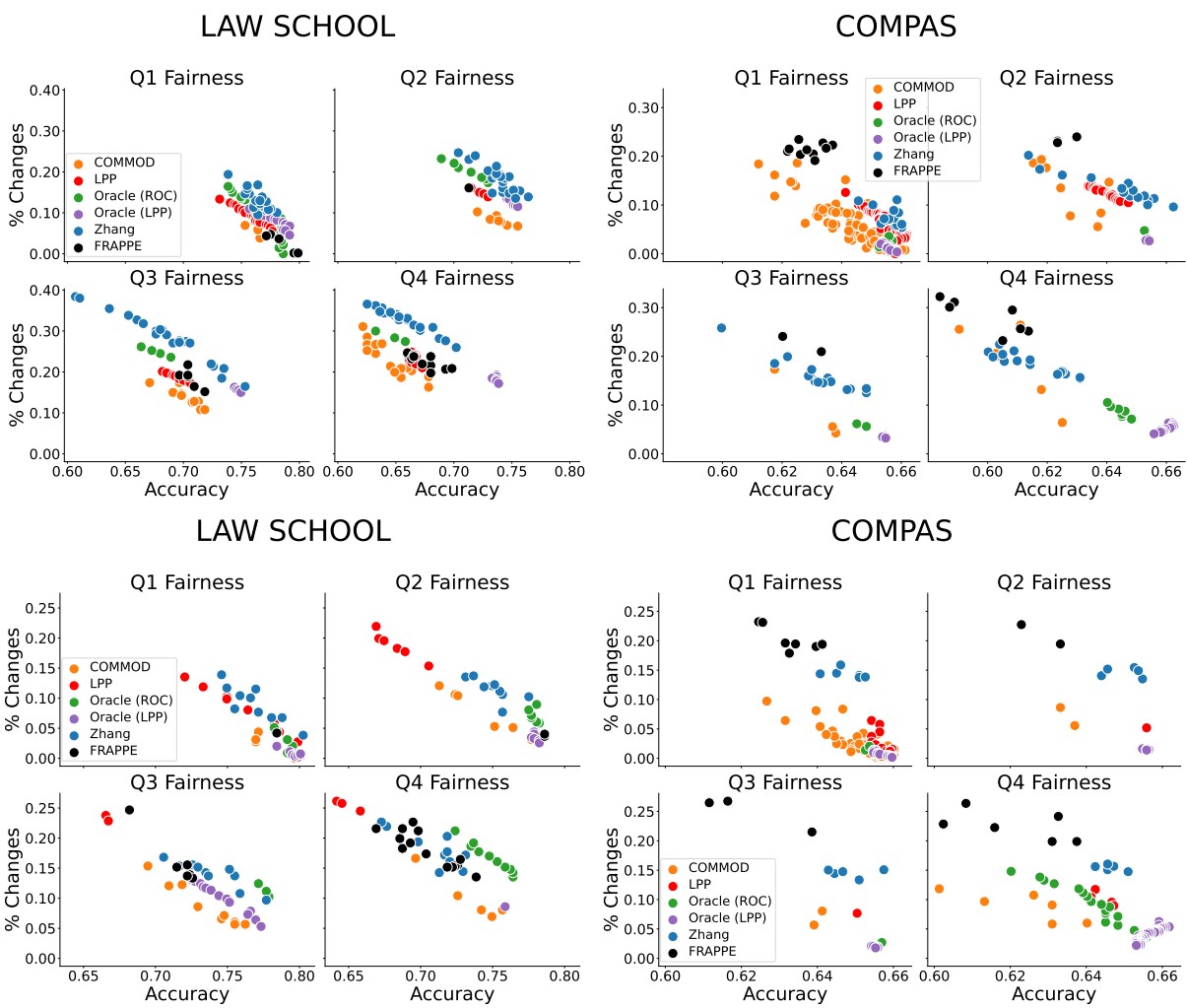

Figure 1: Number of prediction class changes (y-axis) for comparable levels of fairness (P-Rule or DM quartiles Q1-Q4) and accuracy (x-axis) for DP (top) and EO (bottom) on Law School and Compas.

baselines in this section, we report performance using accuracy, that is a *discrete* metric. Indeed, several competing methods intervene by directly flipping labels, so continuous measures (e.g., calibration error) would not place all approaches on the same footing. Nevertheless, to verify that COMMOD's probability scores remain well-behaved, we also analyze their calibration (cf. Figure 9 in the Appendix). As $\mathcal{P}$ is directly linked to the fairness and accuracy levels of the models, we intend to evaluate $\mathcal{P}$ for models with *comparable* levels of fairness and accuracy. To do so, we discretize the fairness scores into four segments, defined as quartiles of the scores of *AdvDebias*: Q1 corresponds to the most biased models, and Q4 to the most fair ones. The full details of these segments are available in Appendix C.3, along with a robustness analysis (cf. Appendix D.1) in which, to ensure that our results do not depend on the specific segment definition chosen, we reproduce the same experiment with other segment definitions. For each segment, we then display the Pareto graph between $\mathcal{P}$ (y-axis, lower is better) and Accuracy (x-axis, higher is better) in Figure 1: in this representation, the most efficient method will be the closest one to the bottom left corner. An aggregated view of these results can be found in Table 1.

We observe that for similar levels of accuracy and fairness, COMMOD consistently achieves lower values of number of changes than its competitors. For less fair models (segment Q1), the difference is more subtle: as models put less emphasis on fairness than on accuracy, they tend to adopt a similar behavior, i.e. the one of the biased model $f$. In other segments however, the value of COMMOD is more notable.

### 6.3 Experiment 2: interpretability of the model updates

In this second experiment, we aim to evaluate the quality of the explanations generated with COMMOD.

#### 6.3.1 Illustrative results

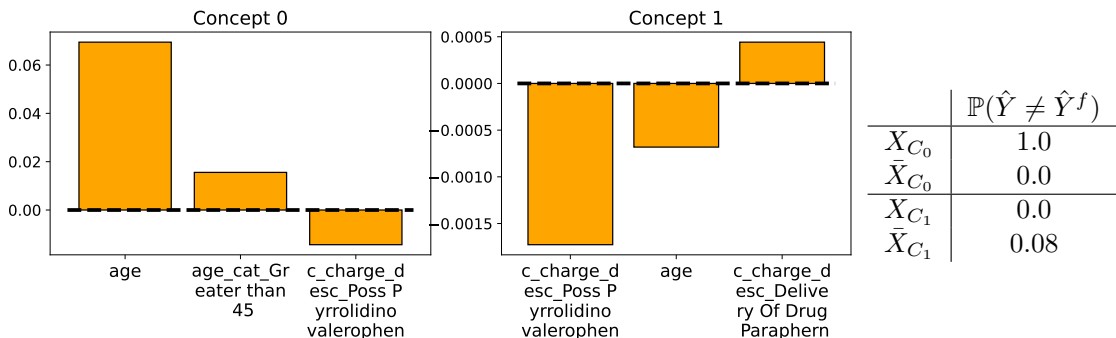

Figure 2: Example of explanation for Compas. Left: feature contributions to the concepts. Right: $\mathbb{P}(\hat{Y} \neq \hat{Y}^f)$ for the segments described by these features.

Starting with an example of explanation on the Compas Dataset. After training COMMOD with $k = 2$ concepts, we obtain a final test P-Rule score of 0.78 (up from 0.60 with $f$) and accuracy of 0.62 (down from 0.66) with $\mathcal{P} = 0.08$. The first two plots of Figure 2 highlight the diversity between these concepts, as they target different features. For instance, the concept $C_0$, contributing positively ($w^0 = 0.41$) to class changes, is primarily activated by older individuals; on the other hand, $C_1$, contributing negatively ($w^1 = -0.23$), targets a specific crime category. In the third graph, we calculate the probability values $\mathbb{P}(\hat{Y} \neq \hat{Y}^f)$ in the sets corresponding to the instances activated by the corresponding features. Although preliminary, these observations show the potential of COMMOD to explain the debiasing process.

#### 6.3.2 Quantitative results

**Sparsity and diversity.** We now aim to generalize the previous observation by verifying that the concepts learned are diverse and sparse over the features of $\mathcal{X}$. To assess sparsity, we evaluate the total concept sparsity computed by summing across concepts the sparsity of each matrix after applying a threshold of $\epsilon = 0.01$: $\sum_{i \leq k} ||W^i > \epsilon||_0$. For diversity, we measure the Jaccard Index between the signed non-null coefficients: diverse concepts may still use the same feature of $X$ if said feature contribution is in the opposite direction. As a baseline, we compare COMMOD (with $k = 2$) with FRAPPE (Ţifrea et al., 2023), when the latter is trained using a linear network with the same architecture as the one used for our ratio $r_{w_g}$ for their additive term. Figure 3, highlights that for all levels of number of changes, the concepts learned using COMMOD are indeed more sparse and diverse as expected.

**Locality.** Finally, we aim to show that the instances targeted by COMMOD are generally located in regions that are easier to interpret. Concretely, the objective of this evaluation is to assess whether COMMOD enforces fairness by targetting bigger regions of the feature space rather than performing scattered individual changes, which would be more challenging to understand for a user. In classification, there exists a theoretical understanding on how a class of models is interpretable based on how it can be translated into a decision tree (Bressan et al., 2024). Driven by this work, we propose a new evaluation protocol relying on measuring the accuracy of a decision tree of constrained depth trained to predict whether an instance has its prediction changed or not (i.e. on labels $\mathbf{1}_{\hat{Y} \neq \hat{Y}^f}$). The idea behind this evaluation protocol is that a decision tree with a constrained depth will more accurately model prediction changes if these changes can be described with a lower number of features, and if these instances are more concentrated in the feature space. We report in Figure 4 the F1 scores of the decision trees trained to predict the class changes of COMMOD, FRAPPE and *AdvDebias*. We observe that for all values of maximum tree depth considered, thanks to the regularization

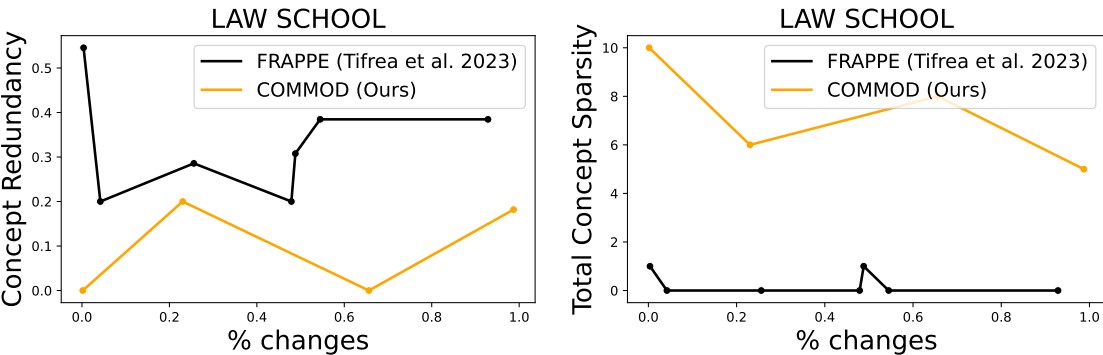

Figure 3: Comparison of concept redundancy (left, lower is better) and sparsity (right, higher is better) for the Law School dataset between FRAPPE and COMMOD.

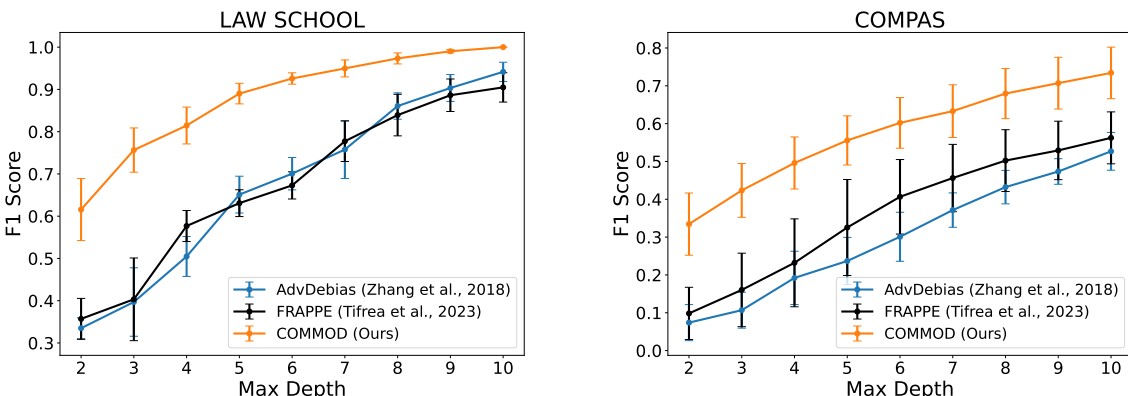

Figure 4: Avg. F1 Score with standard deviations computed over 5 runs for each model (with similar levels of both fairness and accuracy) of a decision tree trained on labels $\mathbf{1}_{\hat{Y} \neq \hat{Y}^f}$ depending on the tree depth.

with $\mathcal{L}_{concept}$, the predictions changed by COMMOD are indeed much easier to interpret with a decision tree.

### 6.4 Sensitivity analysis

Most of the results presented earlier were conducted using one random seed, as concurrently observing the three metrics of interest (accuracy, p-rule and number of changes) across different seeds is difficult to visualize. Indeed, changing the seed of the experiments results in different train-test data splits, and in a different biased model $f$ in the end. Ultimately, this impacts the results of COMMOD, as the accuracy reached by the biased model defines an upper bound for all the other models (disregarding the models complexity). We therefore aim to confirm that the performance of COMMOD are majoritarely driven by the performance of the biased model and the values chosen for COMMOD's hyperparameters.

To do so, we replicate with 20 different seeds the experimental protocol described in the paper: splitting the data, training a biased model, and training COMMOD with various parameters (we set $\lambda_{fair}$ to values varying between 0.5 and 1.5, and $\lambda_{ratio}$ to values varying between 0.01 and 0.1). We then model the relationship between the performance of COMMOD and the performance of the biased model and COMMOD's hyperparameters using three linear regression models:

$$\text{dist}(f, g) \sim \beta_0 + \beta_1\text{fair}_f + \beta_2\text{acc}_f + \beta_3\lambda_{ratio} + \beta_4\lambda_{fair}$$
$$\text{Acc}_g \sim \beta_0 + \beta_1\text{fair}_f + \beta_2\text{acc}_f + \beta_3\lambda_{ratio} + \beta_4\lambda_{fair} \qquad (6)$$
$$\text{fair}_g \sim \beta_0 + \beta_1\text{fair}_f + \beta_2\text{acc}_f + \beta_3\lambda_{ratio} + \beta_4\lambda_{fair}$$

With dist, fair and Acc being defined in Section 3. For the three regressions, the resulting coefficients of determination $R^2$ are 0.96, 0.99 and 0.97. This shows that the performance of COMMOD along the three considered dimensions almost exclusively depends on the performance of the initial model $f$, as well as on COMMOD's hyperparameters. The impact of other sources of possible stochasticity, e.g. due to the initialization of the neural network, remains therefore limited, thus confirming the stability of our method.

## 7 Conclusion

In this work, we introduced COMMOD, a novel model update technique that manages to enforce fairness and accuracy while making less, and more interpretable, changes to the original biased classifier. Additionally, we also provided theoretical results for this new optimization problem. Future works include researching the Controlled Model Debiasing task in non-linear settings, and further exploring the tradeoffs between the different hyperparameters to propose automated selection strategies.

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

# A  Overview of post-processing approaches

In order to better have an overview of existing post-processing approaches, we decided to summarize (part of) them into the following table.

| Paper | Model agnostic | No Sensitive at test time | Metric optimized | Minimizes changes |
|---|---|---|---|---|
| MNB (Calders & Verwer, 2010) | ✗ (NB) | ✗ | DP | ✗ |
| Leaf Relabeling (Kamiran et al., 2010) | ✗ (C4.5) | ✗ | DP | ✗ |
| ROC (Kamiran et al., 2012) | ✓ | ✗ | DP | ✗ |
| EO post-processing (Hardt et al., 2016) | ✓ | ✗ | EO | ✗ |
| Information Withholding (Pleiss et al., 2017) | ✓ | ✗ | EO | ✗ |
| IGD post-processing (Lohia et al., 2019) | ✓ | ✗ | DP | ✗ |
| MultiaccuracyBoost (Kim et al., 2019) | ✓ | ✗ | other | ✗ |
| CFBC (Chzhen et al., 2019) | ✓ | ✗ | EO | ✗ |
| Wass-1 post-processing (Jiang et al., 2020) | ✓ | ✗ | DP | ✓ |
| Wass-1 Penalized LogReg (Jiang et al., 2020) | ✗ (LogReg) | ✓ | DP | ✓ |
| FST (Wei et al., 2020) | ✓ | ✓ | DP, EO | ✗ |
| RNF (Du et al., 2021) | ✗ (NN) | ✓ | DP, EO | ✗ |
| FCGP (Nguyen et al., 2021) | ✓ | ✗ | DP | ✓ |
| FairProjection (Alghamdi et al., 2022) | ✓ | ✗ | DP, EO | ✓ |
| FRAPPÉ (Ţifrea et al., 2023) | ✓ | ✓ | DP, EO | ✓ |
| LPP (sensitive-unaware) (Xian & Zhao, 2024) | ✓ | ✓ | DP, EO | ✗ |
| LPP (sensitive-aware) (Xian & Zhao, 2024) | ✓ | ✗ | DP, EO | ✗ |
| **COMMOD (Ours)** | ✓ | ✓ | DP, EO | ✓ |

Table 2: Comparison between post-processing methods. NB stands for Naive Bayes, and NN for Neural Networks.

From the table above, we can observe that our method is one of the few that is simultaneously model-agnostic, does not require the sensitive variable during inference, and explicitly minimizes the difference between the black-box scores and the edited ones.

# B   Proofs of results in Section 4

In this section, we present the proofs for the propositions and lemmas stated in the paper. Additionally, we offer some context for certain lemmas from Menon & Williamson (2018), which remain necessary within our new framework of Equation 1.

## B.1   Bayes-Optimal Classifier (Result 1)

Building on the results proposed in the paper, we first analyze the Bayes-Optimal Classifier of Equation 1. To facilitate this analysis, we draw upon the work of Menon & Williamson (2018), who demonstrated that by translating the components of the optimization problem into a cost-sensitive framework, the problem becomes analytically equivalent but more tractable to solve.

### B.1.1   A cost-sensitive view of fairness Menon & Williamson (2018).

For any randomized classifier $g \colon \mathcal{X} \times [0,1] \to [0,1]$ that post-process the prediction of a classifier $f \colon \mathcal{X} \to [0,1]$ we can write $FNR(g; \bar{\mathcal{D}}) = P(\hat{Y} = 0 \mid S = 1)$ and $FPR(g; \bar{\mathcal{D}}) = P(\hat{Y} = 1 \mid S = 0)$, where we recall that $\hat{Y} = \mathbf{1}_{g(X) > 0.5}$.

If we want to study for example Demographic Parity, we can refers and compute the following quantity to measure fairness:

$$R_{fair}(g) = DI(g; \bar{\mathcal{D}}) = FPR(g; \bar{\mathcal{D}})/(1 - FNR(g; \bar{\mathcal{D}})). \tag{7}$$

Then, we can translates this quantity into a cost-sensitive risks.

**Lemma B.1** (Lemma 1 in Menon & Williamson (2018))**.** *Pick any random classifier g. Then, for any* $\tau \in (0, \infty)$*, if* $k = \frac{\tau}{1+\tau}$

$$DI(g; \bar{\mathcal{D}}) \geq \tau \Leftrightarrow CS_{bal}(g; \bar{\mathcal{D}}, 1 - k) \geq k,$$

*where* $CS_{bal}(g; \bar{\mathcal{D}}, c) = (1 - c) \cdot FNR(g; \bar{\mathcal{D}}) + c \cdot FPR(g; \bar{\mathcal{D}})$*.*
*Further, we can also rewrite the desired "symmetrised" fairness constraint as*

$$\min(R_{fair}(g), R_{fair}(1 - g)) \geq \tau \Leftrightarrow CS_{bal}(g; \bar{\mathcal{D}}, \bar{c}) \in [k, 1 - k],$$

*for a suitable choice of* $\bar{c}$*.*

### B.1.2   A cost-sensitive view of minimal changes (Ours).

In the paper, we propose then to find an equivalence of the extra (compared to the usual optimization problem in fairness) constraint we have in our new optimization problem. To achieve this, we present Lemma 4.1, whose proof is provided below.

*Lemma 4.1 of Section 4.1.* By law of total probability we have:

$$P(\hat{Y}^f \neq \hat{Y}) = P(\hat{Y}^f \neq \hat{Y} \mid \hat{Y} = 1)P(\hat{Y} = 1) + P(\hat{Y}^f \neq \hat{Y} \mid \hat{Y} = 0)P(\hat{Y} = 0)$$

Since we are in a binary classification setting, i.e. $\hat{Y}^g \in \{0, 1\}$, we can define

$$c^* = P(\hat{Y} = 1), \quad 1 - c^* = P(\hat{Y} = 0).$$

Hence,

$$P(\hat{Y}^f \neq \hat{Y}) \leq p \Leftrightarrow c^* P(\hat{Y}^f \neq \hat{Y} \mid \hat{Y} = 1) + (1 - c^*)P(\hat{Y}^f \neq \hat{Y} \mid \hat{Y} = 0)P \leq p$$
$$\Leftrightarrow c^* FNR(g; \mathcal{D}^*) + (1 - c^*)FPR(g; \mathcal{D}^*) \leq p$$
$$\Leftrightarrow CS_{bal}(g; \mathcal{D}^*, c^*) \leq p.$$

$\square$

### B.1.3 Bayes-Optimal-Classifier.

Finally, we can rewrite in a Lagrangian version the cost-sensitive problem and solve it analytically.

**Lemma B.2.** *Pick any distributions $\mathcal{D}, \bar{\mathcal{D}}, \mathcal{D}^*$, fairness measure DI and constraint on number of changes. Pick any $c, \tau, p \in (0, 1)$. Then, $\exists \lambda_{fair}, \lambda_{ratio} \in \mathbb{R}, \bar{c}, c^* \in (0, 1)$ with*

$$
\begin{aligned}
\min_g \quad & CS(g; \mathcal{D}, c) \\
s.t. \quad & \min(R_{fair}(g), R_{fair}(1 - g)) \geq \tau \\
& P(\hat{Y}^f \neq \hat{Y}) \leq p
\end{aligned}
\tag{8}
$$

*is equivalent to*

$$
\min_g \quad CS(g; \mathcal{D}, c) - \lambda_{fair} CS_{bal}(g; \bar{\mathcal{D}}, \bar{c}) - \lambda_{ratio} CS_{bal}(g; \mathcal{D}^*, c^*).
$$

*Proof (Lemma B.2).* By Lemma B.1 and Lemma 4.1 we have that

$$
\min(R_{fair}(g), R_{fair}(1 - g)) \geq \tau \Leftrightarrow CS_{bal}(g; \bar{\mathcal{D}}, \bar{c}) \in [k, 1 - k],
$$
$$
P(\hat{Y}^f \neq \hat{Y}) \leq p \Leftrightarrow CS_{bal}(f; \mathcal{D}^*, c^*) \leq p.
$$

Consequently, for $\lambda_1, \lambda_2, \lambda_3 \geq 0$, the corresponding Lagrangian version will be

$$
\min_g CS(g; \mathcal{D}, c) + \lambda_1 (CS_{bal}(g; \bar{\mathcal{D}}, \bar{c}) - (1 - k)) - \lambda_2 (CS_{bal}(g; \bar{\mathcal{D}}, \bar{c}) - k) - \lambda_3 (CS_{bal}(g; \mathcal{D}^*, c^*) - p).
$$

Letting $\lambda_{fair} = \lambda_1 - \lambda_2$ and $\lambda_{ratio} = \lambda_3$ shows the results. □

**Lemma B.3.** *Pick any randomised classifier $g$. Then, for any cost parameter $c \in (0, 1)$,*

$$
CS(g; c) = (1 - c)\pi + \mathbb{E}_X[(c - \eta(X))g(X)],
$$

*where $\eta(x) = P(Y = 1 | X = x)$ and $\pi = P(Y = 1)$*

*Proof (Lemma B.3).* For the case of accuracy and fairness constraints the proof is in Lemma 9 of Menon & Williamson (2018). For what concerns the constraint on similarity between the black-box predictions $\hat{Y}^f$ and the one coming from the post-processed classifier $g(X)$, FPR and FNR refers to $\hat{Y}^f$ and no more to $S$. Hence, in the same spirit of the above mentioned proofs,

$$
\begin{aligned}
CS(g; \mathcal{D}^*, c^*) &= (1 - c^*)\pi^* \mathbb{E}_{X | \hat{Y}^f = 1}[1 - g(X)] + c^*(1 - \pi^*)\mathbb{E}_{X | \hat{Y}^f = 0}[g(X)] \\
&= (1 - c^*)\pi^* \mathbb{E}_X[(c^* - \eta^*(X))g(X)],
\end{aligned}
$$

where $\eta^*(X) = P(\hat{Y}^f = 1 | X = x)$ and $\pi^* = P(\hat{Y}^f = 1)$. □

After introduced all these lemma and definitions, we are now able to provide the proof of Lemma 4.2 stated in the paper and that provides the definition of the Bayes-Optimal Classifier.

*Proof (Lemma 4.2).* By Lemma B.3, measures of accuracy, fairness and minimal changes are

$$
CS = (1 - c)\pi + \mathbb{E}_X[(c - \eta(X))g(X)],
$$
$$
CS_{bal}^{fair} = (1 - \bar{c})\bar{\pi} + \mathbb{E}_X[(\bar{c} - \bar{\eta}(X))g(X)],
$$
$$
CS_{bal}^{ratio} = (1 - c^*)\pi^* + \mathbb{E}_X[(c^* - \eta^*(X))g(X)].
$$

Then, ignoring constants not dependent from $g$, the overall objective becomes

$$
\begin{aligned}
\min_g R_{CS}(g; \mathcal{D}, \bar{\mathcal{D}}, \mathcal{D}^*) &= \min_g CS(g; \mathcal{D}, c) - \lambda_{fair} CS_{bal}(g; \bar{\mathcal{D}}, \bar{c}) - \lambda_{ratio} CS_{bal}(g; \mathcal{D}^*, c^*) \\
&= \min_g \mathbb{E}_X[\{\eta(x) - c - \lambda_{ratio}(\bar{\eta}(x) - \bar{c}) - \lambda_{fair}(\eta^*(x) - c^*)\}g(X)] \\
&= \min_g \mathbb{E}_X[-s^*(X)g(X)].
\end{aligned}
$$

Thus, at optimality, when $s^*(x) \neq 0$, $g_{opt} = [\![s^*(x) > 0]\!]$. When $s^*(x) = 0$, any choice of $g_{opt}$ is admissible. □

**B.2   Fairness Level Under a Maximum Change Constraint (Result 2)**

*Proof of Proposition 4.4.* Without loss of generality, let us suppose that $P\text{-}Rule(f) = C \cdot \frac{\gamma_{11}}{\gamma_{10}}$. Then, by doing $K$ changes with $K < K_s$, we get a new value of $P\text{-}Rule'$ that is given by

$$P\text{-}Rule^\alpha(g) = C \cdot \frac{\gamma_{11}(f) + \alpha}{\gamma_{10}(f) - (K - \alpha)},$$

Further, if we compute the derivative with respect to alpha of $P\text{-}Rule'$ we get

$$\frac{d}{d\alpha} P\text{-}Rule^\alpha(g) = C \frac{\gamma_{11}(f) - K - \gamma_{10}(f)}{(\gamma_{10}(f) - (K - \alpha))^2}, \tag{9}$$

and we observe that $P\text{-}Rule^\alpha(g)$ is either everywhere decreasing or increasing. Hence,

$$\max_{0 \le \alpha \le K} P\text{-}Rule^\alpha(g) = \begin{cases} P\text{-}Rule^0(g) & \text{if } K > \gamma_{10}(f) - \gamma_{11}(f) \\ P\text{-}Rule^K(g) & \text{otherwise} \end{cases}.$$

Finally, the same reasoning applies to the case where the starting value of the $P\text{-}Rule^\alpha(f)$ is $\frac{1}{C} \frac{\gamma_{11}(f)}{\gamma_{10}(f)}$.   □

*Proof of Proposition 4.5.* Writing out the sum,

$$\text{DM}(x) = \Delta_{\text{TPR}} - \gamma x + \Delta_{\text{FPR}} - \delta(K - x) = \big(\Delta_{\text{TPR}} + \Delta_{\text{FPR}} - \delta K\big) + (\delta - \gamma)\, x.$$

Thus $\text{DM}(x)$ is an affine (linear) function of $x$ with slope $\delta - \gamma$.

- If $\delta - \gamma > 0$, the slope is positive, so $\text{DM}_{\text{sum}}(x)$ is increasing in $x$ and its minimum on $[0, K]$ occurs at $x = 0$ (all flips to FPR-adjustment).

- If $\delta - \gamma < 0$, the slope is negative, so $\text{DM}_{\text{sum}}(x)$ is decreasing in $x$ and its minimum occurs at $x = K$ (all flips to TPR-adjustment).

- If $\delta = \gamma$, the function is constant and any allocation $x \in [0, K]$ yields the same DM.

In all cases, an "extreme" allocation—putting all $K$ flips into the single most effective action—minimizes the sum-based Disparate Mistreatment.   □

## C   Implementation details

### C.1   COMMOD Algorithm

In this section we give a more detailed walk-through of our end-to-end training procedure for COMMOD, as well as a compact pseudocode listing.

**Discussion.**   At each training step, for a minibatch of examples $(x_i, y_i, s_i)$ we first convert the biased model's output probability $f(x_i)$ into logits

$$f_{\text{logit}}(x_i) \;=\; \log\big(f(x_i)/(1 - f(x_i))\big).$$

We then concatenate (or otherwise condition on) both $x_i$ and $f_{\text{logit}}(x_i)$ as input to a small rescaling network $r_{w_g}$, which produces a scalar multiplier $r(x_i)$. The corrected score

$$g(x_i) \;=\; r(x_i)\, f_{\text{logit}}(x_i)$$

is fed both to the final classification loss $\mathcal{L}_Y$ (against $y_i$) and to an adversary $h$ that predicts the sensitive attribute $s_i$, giving $\mathcal{L}_S$. In addition we regularize $r$ to stay close to 1 via

$$\mathcal{L}_{\text{ratio}}(r(x_i)) \;=\; \big\|r(x_i) - 1\big\|^2,$$

and add a concept-based penalty $\mathcal{L}_{\text{concepts}}$. All four terms are weighted and summed, and we backpropagate through $r$ and $h$ jointly.

---

**Algorithm 1** COMMOD Training (simplified pseudocode)

---

**Require:** Pretrained biased model $f$, rescaler $r$, adversary $h$, data $\{(x_i, y_i, s_i, \hat{y}_i)\}$. $\{\hat{y}_i = f(x_i)$ are probabilities, $y_i \in \{0, 1\}$ are true labels$\}$.

1: **for** each epoch **do**
2:     **for** each minibatch $(x, y, s, \hat{y})$ **do**
3:         $f_{\text{logit}} \leftarrow \log\big(\hat{y}/(1 - \hat{y})\big)$
4:         $r \leftarrow r(x)$
5:         $g \leftarrow \sigma\big(r \times f_{\text{logit}}\big)$
6:         **(optional)** Warm-up adversary: update $h$ on $(g, s)$
7:         Compute losses:
$$\mathcal{L}_Y, \ \mathcal{L}_S, \ \mathcal{L}_{\text{ratio}}, \ \text{(optional concept losses)}$$

8:         Combine into total loss $\mathcal{L}$ (with weights and epoch gates)
9:         Update parameters of $r$ (and any concept modules) by backpropagating $\nabla\mathcal{L}$
10:    **end for**
11: **end for**

---

## C.2 Loss and Hyperparameters

As with any deep learning-based method, fine-tuning hyperparameters has been crucial for our COMMOD algorithm. Specifically, we adjusted the weights of the terms in the loss function using the hyperparameters $\lambda_{\text{fair}}$, $\lambda_{\text{ratio}}$, and $\lambda_{\text{concepts}}$. To explore the entire Pareto frontier of Fairness vs. Accuracy, as presented in this paper, we performed a grid search over these hyperparameters. For implementation purposes, we occasionally found it easier to decompose the concept loss $\mathcal{L}_{\text{concepts}}$ into the sum of sparsity loss and diversity loss, each controlled by separate hyperparameters. The range of values we tested remained consistent across different datasets, with $\lambda_{\text{fair}} \leq 10$, $\lambda_{\text{ratio}} \leq 0.5$, and $\lambda_{\text{concepts}} \leq 1$.

## C.3 Quartiles definition

In the paper, we referred several times to the quartiles of fairness and accuracy. We used them in order to compare the methods in the region of the Pareto for which they have approximately the same levels of fairness and accuracy. These quartiles have been computed on the *AdvDebias* method and their values are shown in the tables below.

## C.4 Law School Dataset

### C.4.1 Demographic Parity.

We recall that in order to measure Demographic Parity we used the *P-Rule* (the higher, the better). The value of the point ($fairness$, $accuracy$) of the black-box model (Logistic Regression) is (0.2764, 0.7970).

| Quartile | Fairness range | Accuracy range |
|----------|----------------|----------------|
| Q1 | [0, 0.5587) | [0, 0.6709) |
| Q2 | [0.5587, 0.7212) | [0.6709, 0.7294) |
| Q3 | [0.7212, 0.8719) | [0.7294, 0.7550) |
| Q4 | [0.8719, 1] | [0.7550, 1] |

Table 3: Fairness (Demographic Parity) and Accuracy Quartiles on Law School dataset.

### C.4.2 Equalizing Odds.

We recall that in order to measure Equalizing Odds we used the *Disparate Mistreatment* (the lower, the better). The value of the point ($fairness$, $accuracy$) of the black-box model (Logistic Regression) is (0.4935, 0.7970).

| Quartile | Fairness range | Accuracy range |
|---|---|---|
| Q1 | (0.3429, 1] | [0, 0.7230) |
| Q2 | (0.2415, 0.3429] | [0.7230, 0.7458) |
| Q3 | (0.1503, 0.2415] | [0.7458, 0.7577) |
| Q4 | [0, 0.1503] | [0.7577, 1] |

Table 4: Fairness (Equalizing Odds) and Accuracy Quartiles on Law School dataset.

## C.5 COMPAS Dataset

### C.5.1 Demographic Parity

. We recall that in order to measure Demographic Parity we used the *P-Rule* (the higher, the better). The value of the point ($fairness$, $accuracy$) of the black-box model (Logistic Regression) is (0.6310, 0.6580).

| Quartile | Fairness range | Accuracy range |
|---|---|---|
| Q1 | [0, 0.7345) | [0, 0.6242) |
| Q2 | [0.7345, 0.7695) | [0.6242, 0.6391) |
| Q3 | [0.7695, 0.8058) | [0.6391, 0.6537) |
| Q4 | [0.8058, 1] | [0.6537, 1] |

Table 5: Fairness (Demographic Parity) and Accuracy Quartiles on COMPAS dataset.

### C.5.2 Equalizing Odds.

We recall that in order to measure Equalizing Odds we used the *Disparate Mistreatment* (the lower, the better). The value of the point ($fairness$, $accuracy$) of the black-box model (Logistic Regression) is (0.2828, 0.6580).

| Quartile | Fairness range | Accuracy range |
|---|---|---|
| Q1 | (0.2198, 1] | [0, 0.6242) |
| Q2 | (0.2079, 0.2198] | [0.6242, 0.6391) |
| Q3 | (0.1869, 0.2079] | [0.6391, 0.6537) |
| Q4 | [0, 0.1869] | [0.6537, 1] |

Table 6: Fairness (Equalizing Odds) and Accuracy Quartiles on COMPAS dataset.

# D   Additional Experiments and Further Results

## D.1   Experiment 1: robustness over segment definition

To ensure that the results of Experiment 1 are robust to the choice of the segment definitions for accuracy and fairness, we propose in this section to reproduce the experiment with other segment definitions. To define these segments, we thus propose, to take the quartile values of the results achieved by LPP (Definition 2) and ROC (Definition 3), instead of AdvDebias.

### D.1.1   Definition 2.

The quartiles values obtained through the result of LPP are the following

| Quartile | Fairness range | Accuracy range |
|----------|----------------|----------------|
| Q1 | [0, 0.4709] | [0, 0.7459) |
| Q2 | (0.4709, 0.5918] | [0.7459, 0.7541) |
| Q3 | (0.5918, 0.7763] | [0.7541, 0.7723) |
| Q4 | [0.7763, 1.0] | [0.7723, 1] |

Table 7: Fairness (DP) and Accuracy Quartiles (Definition 2) on Law School dataset.

Using this values, we can then plot the analogue of Fig. 1

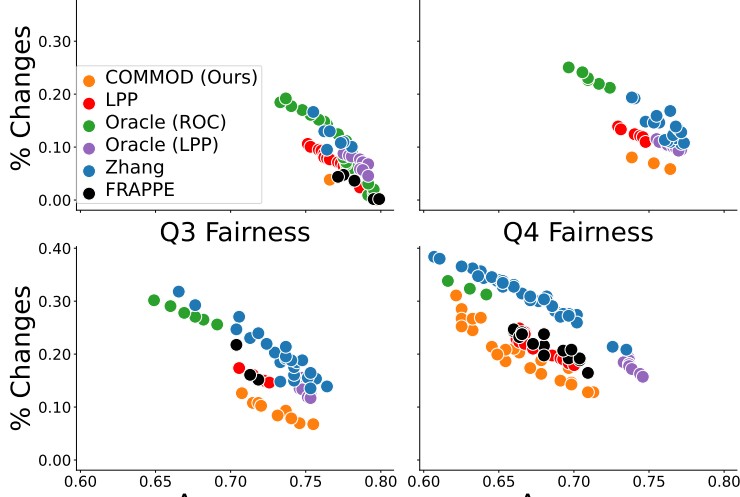

Figure 5: Experiment 1 results, with Definition 2 for Accuracy and Fairness segments

### D.1.2   Definition 3.

The quartiles values obtained through the results of ROC are the following
Using this values, we can then plot the analogue of Fig. 1

## D.2   Accuracy and Fairness Performance

As traditionally done for bias mitigation methods, we evaluate the efficacy of COMMOD to preserve accuracy when enforcing fairness through the so-called Pareto plot. Figure 7 shows the accuracy and fairness scores

| Quartile | Fairness range | Accuracy range |
|----------|----------------|----------------|
| Q1 | [0, 0.3247] | [0, 0.7033) |
| Q2 | (0.3247, 0.4127] | [0.7033, 0.7559) |
| Q3 | (0.4127, 0.5695] | [0.7559, 0.7793) |
| Q4 | [0.5695, 1] | [0.7793, 1] |

Table 8: Fairness (DP) and Accuracy Quartiles (Definition 3) on Law School dataset.

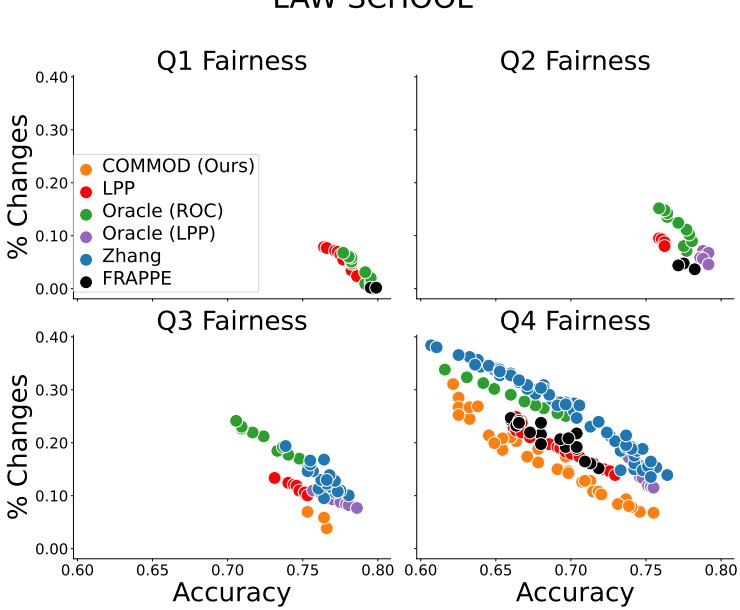

Figure 6: Experiment 1 results, with Definition 3 for Accuracy and Fairness segments

achieved by all methods on Law School and Compas datasets (each dot represents one run). In terms of direct competitors, we observe that COMMOD outperforms LPP (resp. FRAPPE) on the Law School dataset (resp. Compas) and achieves similar results in the Compas dataset (resp. Law School). Further, we show here that even by adding the Minimal Changes and Interpretability constraints in Equation 4 COMMOD achieves almost similar results on all datasets as the in-processing method *AdvDebias*. Finally, as expected, *Oracle* (and also by ROC in Compas) generally outperform all methods. Hence, COMMOD remains a competitive algorithm in terms of fairness and accuracy, regardless of its other benefits, addressed in the next sections.

### D.3   Intuition behind linear self-explainable model.

One might question why we chose to introduce explainability into our method through a linear architecture that learns concepts as linear combinations of the input features. Typically, using a linear model instead of a more complex one can lead to a decrease in model performance. However, the rationale behind our choice is clarified through the following experiment, where we evaluate different architectures for the network that learns the multiplicative ratio. From the plot above, we can observe that the linear architecture performs similarly to more complex models. This observation aligns with the findings in Ţifrea et al. (2023), where a similar conclusion was drawn regarding their additive term, which is conceptually similar to our multiplicative ratio. Based on these insights, we decided to design the architecture of the ratio as a one-hidden-layer linear network. In this design, the hidden layer acts as a bottleneck that represents the concepts, and the output layer computes the ratio as a linear combination of these concepts.

Additionally, maintaining linearity between the concepts and the output ratio allows us to easily determine

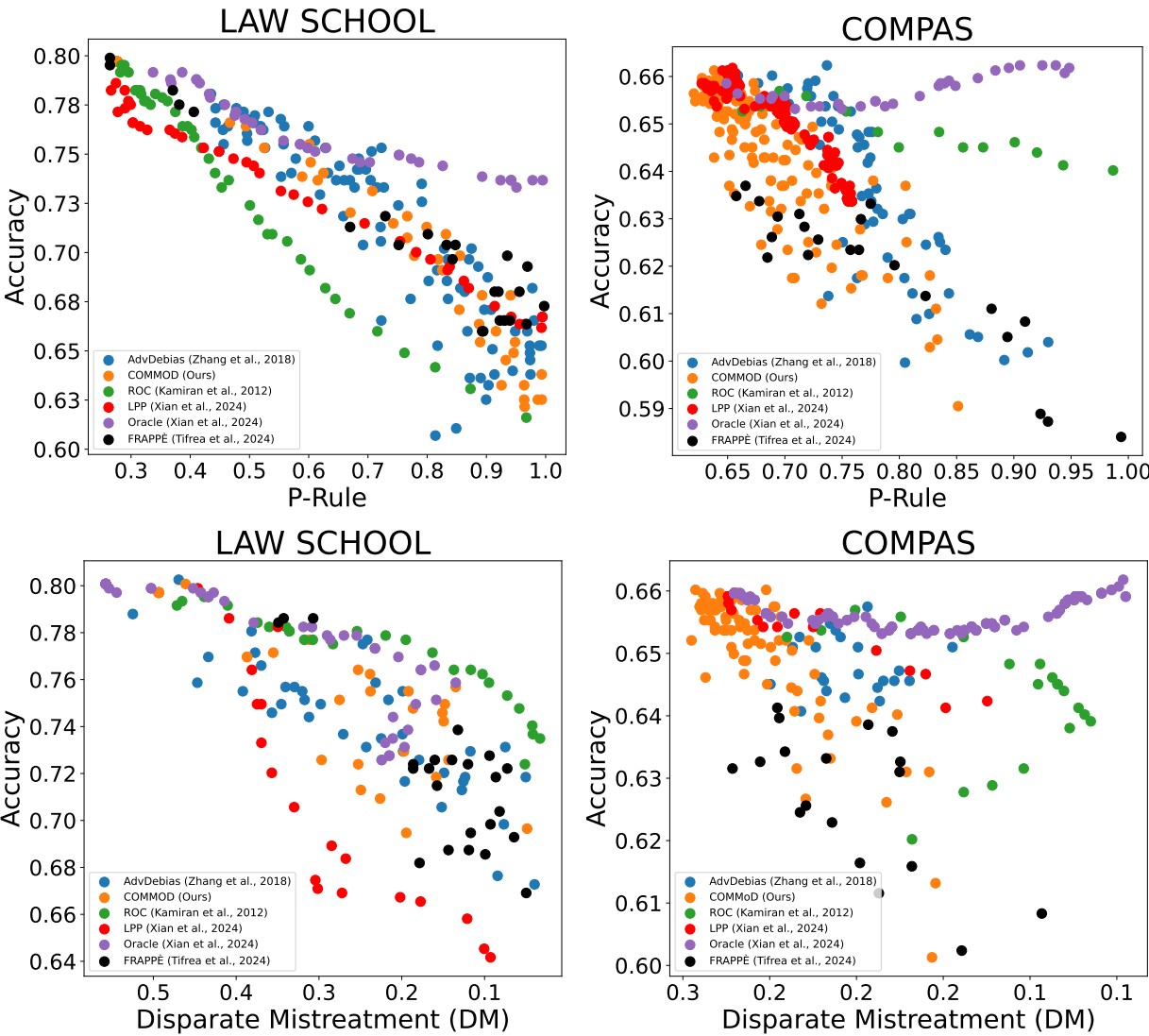

Figure 7: Fairness (x-axis) vs Accuracy (y-axis) trade-offs on Law School and Compas datasets for Demographic Parity (top) and Equalized Odds (bottom).

the direction (positive or negative) in which each concept influences the ratio's value. This design choice enhances both the interpretability and transparency of the model.

### D.4 Benefits of MSE Loss

In this section, we propose a visualization in support to the MSE loss we chose to keep as similar as possible the edited scores with the ones outputted by the black-box algorithm. To do so, we used a "calibration" plot on edited scores vs black-box scores for our COMMOD algorithm and for *AdvDebias* for similar levels of (*fairness*, *accuracy*). From such a plot, one can observe that the curve of COMMOD is more stick to the dashed line, representing when probabilities are not changed at all by the editing. On the other hand, we observe that in *AdvDebias* even if we do not change a prediction label, the probabilities are more scattered in the area of the dashed line. This behavior is beneficial for our motivation. When you want to edit a model for fairness, you usually want to do changes that flip labels (and hence that modifies the values of your fairness definition) and keep the other instances at the same confidence they were before.

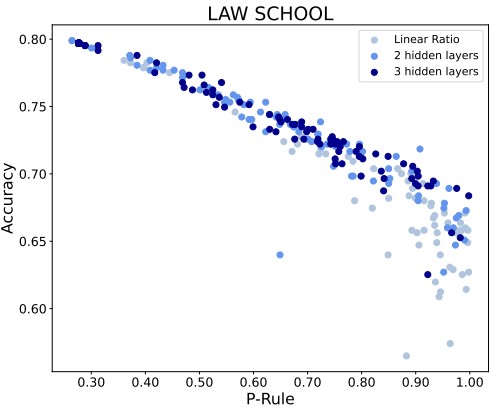

Figure 8: Fairness vs Accuracy trade-off on Law School for different ratio architectures. Each dot corresponds to one model run.

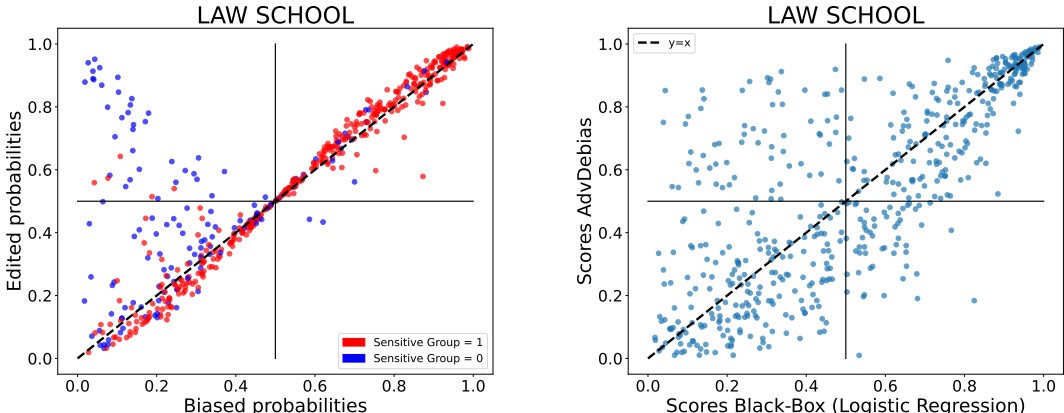

Figure 9: Calibration plot (predicted probabilities vs black-box probabilities) on Law School dataset for Demographic Parity.

### D.5   Posthoc vs self explainable

As a final experiment, we aim to demonstrate the need for an approach specifically designed for Controlled Model Debiasing such as COMMOD. For this purpose, we build on the previous experiment and show that the combination of FRAPPE (which minimizes the number of changes) and a post-hoc interpretability tool leads to less efficient models in terms of debiasing; in other words, that adding interpretability in a post-hoc way comes at a higher cost for fairness. We train FRAPPE and COMMOD on the Law School dataset such that their scores in terms of accuracy, fairness and number of changes are comparable. We then train a decision tree to model FRAPPE's additive output, and show in Figure 10 the final P-Rule obtained by COMMOD and this built competitor. We observe that, regardless of the tree's depth, controlling the debiasing by introducing interpretability in a post-hoc way heavily degrades fairness, contrary to COMMOD.

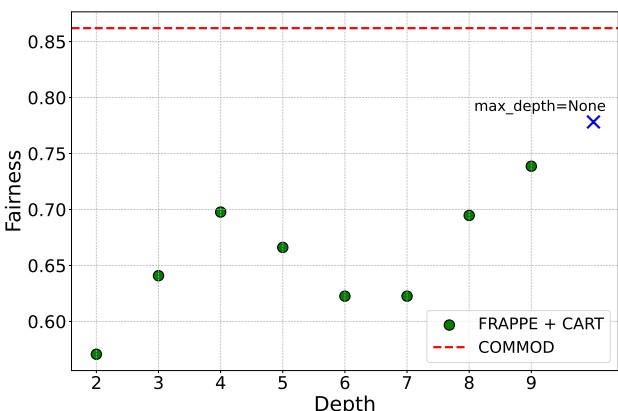

Figure 10: Comparison of fairness levels of FRAPPE+CART (trained on the additive term) with the one of a self-explainable model for different levels of depth

