# OpenReview forum: "Controlled Model Debiasing through Minimal and Interpretable Updates"
_TMLR — Accepted by TMLR_

### Review · Reviewer_7L1a · 2025-03-24

**Summary Of Contributions:**

This paper tackles the challenge of postprocessing a model with the goal of increasing its fairness. The approach is formally a supervised learning task which relies on two insights: The differences between new fair model and existing one should be (1) interpretable and (2) minimal. The paper introduces a novel algorithm, COMMOD, which is model-agnostic, does not require sensitive attribute at test time, and makes minimal and interpretable changes between biased and debiased predictions. Experimentally, COMMOD achieves comparable performance to state-of-the-art debiasing methods.

**Audience:**

No

**Claims And Evidence:**

No

**Requested Changes:**

See weaknesses.

**Strengths And Weaknesses:**

### Main Strengths:

- The problem of making classifiers more fair is relevant, especial in some particular application;

- The paper since the very beginning states its main two intuitions, namely that a postprocessed classifier should be similar to the initial model (i.e., minimal changes) and the differences should be easy to explain;

- Using the decision tree to explain the differences is smart and intuitively interesting. Great to have in the paper and I would give more visibility to it, instead of including it as final concept.


### Main Weaknesses:

- The minimal update in Eq 1 is based on the accuracy metric, which essentially relies on thresholding some scores/probabilities at 0.5 to obtain hard predictions. However, the model is based on logistic regression, which makes probabilities (and are also used in Ea 3). Because it's a threshold-based metric, accuracy does not account for small changes in probabilities, which might be the effect of making a model more fair. Wouldn't a metric that accounts for probabilities (e.g., calibration metrics, ranking-based metrics) more suitable for the task?

- Section 5.3 explains the final construction of the model, i.e. how to setup the optimization problem to learn. I find this section very disconnected from the previous ones, and I am not sure how the previous paragraphs fit into the final model. For example, Eq. 1 and 2? How about the explainability of the difference? This part introduces the loss function, which is not considered in most of the paper.

- Eq. 3 uses logits but how to estimate them is unclear; Similarly, r(X) is modeled as a Neural Network but it's unclear (1) what type of NN, (2) what's the target value, (3) how the network is trained;

- Sec 5.1 mentions the need for a penalization term, which measures the distance between two distributions. The paper proposes a simple L2 norm, which is obviously defined over a dot product. I wonder what's the utility of defining L_ratio with the norm M, if eventually the only norm used is the L2. This feels just like overgeneralizing.

- The experiments look interesting and results seem promising. However, a couple of points for improvement: 1) Most experiments do not show standard deviations or confidence intervals, and the setup section does not report any repetition of experiments, which makes me conclude that only one iteration is used. Therefore, results might only be good because of random noise; 2) Tables can help visualize the superiority of the proposed approach, as they summarize properly the comparison; 3) Figure 4 plots the F1 score, while initially only the accuracy is measured. Why such a change in performance metric? What's the advantage of measuring F1 or Accuracy, especially when using models that produce scores/probabilities?

---

> ### Author Response · Authors · 2025-04-30
> **Answer to reviewer 7L1a**
>
> Thank you for your comments. Please find below our answer to the various weaknesses pointed out:
>
> # On the choice of "Accuracy"
>
> We indeed formalize the difference between the biased model $f$ and the edited model $g$ in a discrete way, as it is the metrics that is the easiest to observe and understand in general. However, we agree that this metrics may hide small probability changes, which might be impactful overall. The current version of the paper proposes a small discussion on that topic in Sec. 5.1, where we introduce a more general metrics for the penalization $\mathcal{L}_{\text{ratio}}$ that allows to account for continuous updates. An additional discussion on the benefit of using a continuous-aware metrics (MSE loss) is also proposed in Appendix D.4, with an experiment showing the impact on the probabilities. Given the reviewer's comment, we propose however to change Eq.1 with a more general formulation, allowing for both discrete and continuous (probability) prediction changes to be observed. The resulting changes can be seen in the paper (Eq. 1 + text in Sec. 3.2).
>
> # On the overall articulation of the paper (weakness 2)
> Section 5.3 describes the final model used to address the presented problem. Equation 4 is thus a direct transcription of Equation 1 using the various loss functions earlier introduced: $\mathcal{L}_Y$  in the first paragraph of Section 5, $\mathcal{L}_S$  in Section 5.3, and $\mathcal{L}_{\text{ratio}}$ in Section 5.1. As discussed, we then use as the final loss function the Lagrangian formulation of this problem, and add a penalization term for the interpretability of the difference. Following the reviewer's comment, we have clarified the discussions and notations in Section 5 overall. (text in red)
>
> # On Equation 3 and overall presentation of the method
> The logits used as inputs by our method are directly calculated using the probabilities $p$ of $f$, using the formula: $\text{logit} = \text{log}(\frac{p}{1-p})$. These logits are then rescaled using amultiplicative factor $r$ the following way:
> - A fully connected neural network outputs a continous value $r(x)$.
> - The target value of the overall architecture is the label $Y$. Concretely, after outputting $r(x)$, this value is multiplied to the biased model prediction $f_{\text{logit}}(x)$: and the loss between $y$ and $r(x) f_{\text{logit}}(x)$ is then calculated.
> - The training of the overall model can be described through the following steps. Given an input $x$, and the corresponding probability $f(x)$:
>     1. Calculate the logit $f_{\text{logit}}(x)$;
>      2. Pass $x$ and $f_{\text{logit}}(x)$ through the network $r$ to obtain $r(x)$;
>     3. Calculate g(x) = $r(x)f_{\text{logit}}(x)$;
>     4. Pass $g(x)$ through the adversarial network $h$;
>     5. Calculate the losses $\mathcal{L}_Y(g(x), Y), \mathcal{L}_S(h(g(x)),S), \mathcal{L}_ratio(r(x))=||r(x)-1||_2$ and $\mathcal{L}_Concepts(r(x))$.
> - Perform a backward pass.
>
> We have added a subsection in the appendix C.1, entitled "COMMOD Algorithm", where we discuss in more details this training procedure, as well as add a pseudocode for the algorithm.
>
> # On the choice of the L2 norm
> While experiments have been conducted using the $L_2$ norm, we believe that the framework benefits from being more general, as choosing other distance metrics would allow to cover other interesting notions of model similarities. These could include for instance using the KL divergence or Wasserstein distance, traditionally used by existing post-processing fairness approaches (e.g. (Tifrea et al. 2024). As discussed in our previous answer ("On the choice of Accuracy"), we have modified the discussion in Sec. 5.1 to better reflect these ideas.
>
> # On Experiments
> 1. The seed was indeed fixed for most experiments (e.g. Fig.1). The reason for this is that allowing the seed to change means that the biased blackbox model $f$ changes as well, and observing the results on the 3 main criteria over a same Pareto graph would be impossible. As a result, we have preferred fixing the seed to ensure comparable results. However, this seed was completely set randomly, as the code submitted shows.
>
> 2. We added Table 1 to summarize the results of our method. They propose an aggregated view of the results shown in Fig.1, and show that overall, COMMOD allows to mitigate bias for a given level of accuracy with a reduced number of changes.
>
> 3. Figure 4 uses the F1 score as the data considered in this experiment is highly unbalanced. Contrary to the rest of the paper which considers $Y$ as the label, here, for evaluation purposes, a decision tree is trained to predict label $1_{\hat{Y} = \hat{Y}^f}$. In this setting, this corresponds to a training label ratio of  $\sim 90/10$, justifying our use of the F1 score to better visualize the results.

---

### Review · Reviewer_fDJu · 2025-03-27

**Summary Of Contributions:**

This paper introduces the concept of Controlled Model Debiasing, which aims to update an already-trained, biased model with minimal and interpretable changes to ensure fairness. The authors propose COMMOD, a model-agnostic algorithm that integrates an adversarial framework to mitigate bias and learns sparse, linear “concepts” that clarify how updates from the old model are made. Theoretical guarantees regarding a fairness-aware Bayes-optimal classifier and the trade-off between fairness and the extent of model changes are provided. Experiments on standard datasets (Law School, Compas) demonstrate that COMMOD achieves performance comparable to state-of-the-art debiasing methods but with fewer prediction changes and clearer global explanations of how and why changes occur.

**Audience:**

Yes

**Claims And Evidence:**

Yes

**Requested Changes:**

please refer to the weaknesses.

**Strengths And Weaknesses:**

**Strengths:**

S1. The paper is mostly well-written and well-presented. It’s also addressing an important problem.

S2. The authors provide theoretical guarantees—both a fairness-aware Bayes-optimal classifier result and analysis of how many label changes are needed to improve demographic parity.

S3. The proposed method, COMMOD, has practical properties such as not needing the sensitive feature at the test time and being model agnostic.

**Weaknesses:**

W1. The authors do not fully motivate the importance of the two desiderata they introduce in CMD. Why are they important, what trade-offs would they introduce and what problems would they solve? These are occasionally mentioned throughout the paper, but having them in the introduction would help.

W2. Could the author clarify their contribution to the theoretical guarantees for the minimal update problem? It seems to be mostly from (Menon & Williamson, 2018) and the establishment of the equivalence between the two settings seems very straightforward.

W3. I struggle to make sense of the constraint (\hat{Y^f} \neq \hat{Y}) \leq p. What ‘p’ exactly and why it's taking that range of values? is this a population risk?

W4. Why the theoretical results in section 4.2 are only analyzed for the DP metric. Some discussions or insights on EO would be nice.

W5. The authors can do a better job in introducing some notations and giving them a little intuition, such as D*, c*, K^s_A.

W6. Can the author elaborate on the connection between the penalty term in 5.1 and Eq 1.  Te The L_2 penalty approximates prediction similarity but does not directly control the probability of label flips (p in Eq 1).

W7. My main concern about the experiments is that the results appear based on a single train/test split—no confidence intervals or seed variance are provided.

W8. By assuming the sensitive attribute taking a binary value, both the theoretical guarantees, as well as the experimental results (with the two small datasets) are limiting.

---

> ### Author Response · Authors · 2025-04-30
> **Answer to reviewer fDJu**
>
> Thank you for your comments. You may find our answers to the various weakness pointed out below:
>
> # W1. Motivation for the two desiderata
> We enrich in the introduction the motivations to support our contribution (text in red).
>
> # W2. On the theoretical guarantees and difference with (Menon \& Williamson, 2018)
> The idea of our first theoretical result is indeed inspired from the one in (Menon \& Williamson, 2018). However, in the setting we consider, there is an additional trade-off between three objectives considered (fairness, accuracy, changes), instead of the classical fairness-accuracy tradeoff considerred in their paper.
> Hence, this result aims to show that the new desideratum introduced, i.e. minimal changes, still allows the problem to be written in a cost-sensitive way, regardless of the fairness definition used. Besides, a similar analysis to find the BOC could be carried.
>
> In addition, we also propose a second theoretical result (Sec. 4.2), that is completely new and independant from (Menon \& Williamson, 2018).
>
> # W3. Clarification on the formalization
> We apologize for the confusion, as there was a typo in the previous version of the paper. This constraint should actually state: $P(\hat{Y}\neq \hat{Y}^f)$ (the $P$ was missing). We have now fixed this typo. Following the suggestion of another reviewer, we have also replaced this constraint in Eq. 1 with a more general one, allowing the distance between $f$ and $g$ to take other definitions (e.g. probability changes)
>
> # W4. On theoretical results with EO.
> While only given for DP, extending the theoretical results to EO can be done in the following ways:
> - Result 1 (Sec. 4.1) can be directly adapted, since the additional constraint we bring does not depend on the fairness criteria, and (Menon and Williamson, 2018) also applies to EO. We have added a sentence describing this idea in Section 4.1.
> - Result 2 (Sec. 4.2) can be extended with some calculation that are similar to the DP result. We provide the corresponding result in Section 4.2 (Proposition 4.5), with the proof in Appendix B.2.
>
> # W5. On notations.
> We have clarified the notations to facilitate the reading.
>
> # W6. On the connection between the penalty term and Eq1.
> Indeed, the penalty introduced as an example does not directly control the label flip, but rather the probability similarity. Following another reviewer's comment, we have updated Equation 1 to make the similarity constraint between $f$ and $g$ more general, thus encompassing both discrete (label-based) or continuous (probability-based) similarities. In that context, the L2 penalty considered in the proposed implementation can be seen as one instanciation of the proposed framework. Furthermore, we provide in Appendix D.4 a discussion on the benefits of considering this probability similarity, with empirically showing the effect on the induced probability changes: using the L2 penalty, we were able to reach similar scores in terms of label-flips than when considering a penalty directly optimizing the label class changes, but with less probability consistency overall.
>
> # W7 On the robustness of the results
> See our answer to reviewer 7L1a on the same topic.
>
> # W8. On the sensitive attribute being binary.
> The theoretical guarantees indeed assume the sensitive attribute is binary. While it’s true that a binary-sensitive attribute setting is a simplification, it is also the convention in the fairness literature, since the metrics usually used to compare debiasing algorithms (DM and P-Rule) are defined on this binary framework.

---

### Review · Reviewer_kgV6 · 2025-04-16

**Summary Of Contributions:**

The paper introduces a novel problem setting called Controlled Model Debiasing, which focuses on modifying a previously trained, potentially biased model into a fairer one by making minimal and interpretable changes. The authors propose a model-agnostic algorithm, COMMOD (COncept-based Minimal MOdel Debiasing), which applies a multiplicative update to a model’s prediction logits. COMMOD combines adversarial learning for fairness enforcement with a concept-based interpretable architecture to provide traceable and sparse explanations for prediction changes. The paper provides theoretical results (e.g., a Bayes-optimal classifier under fairness and change constraints), and empirical evaluations on two standard fairness datasets, comparing COMMOD against existing state-of-the-art methods across multiple fairness metrics and interpretability criteria.

**Audience:**

Yes

**Claims And Evidence:**

Yes

**Requested Changes:**

Could the authors comment on the potential of COMMOD in handling successive updates over time?

Could the authors conduct more extensive qualitative analysis, user studies, or grounding in domain knowledge to validate that these concepts offer practically meaningful interpretability?

**Strengths And Weaknesses:**

Strength:
+ The introduction of "Controlled Model Debiasing" as a formal objective is novel, addressing the practical need for minimally invasive updates to deployed models.
+ The paper provides strong theoretical backing, including a fairness-aware Bayes-optimal classifier and an analysis of fairness-improvement bounds under a change constraint.
+ Unlike many fairness approaches, COMMOD does not require access to sensitive features at inference, enhancing applicability.
+ The concept-based architecture with sparsity and diversity regularization makes COMMOD’s decision changes explainable.

Weakness:
- While Controlled Model Debiasing presents a valuable and less-explored idea, its proposed solution, COMMOD, is very similar to existing fairness methods—especially those that adjust model outputs to meet fairness goals. The use of a multiplicative factor to adjust logits can be seen as a nuanced variation of established post-hoc calibration or output perturbation techniques, which makes its originality questionable. Additionally, the paper does not address how COMMOD would handle successive updates over time—an important consideration in practical deployment where fairness constraints or data distributions may evolve. If one were to apply COMMOD iteratively (e.g., to enforce new or evolving fairness standards), the multiplicative update mechanism may compound and unintentionally drift significantly from the original model's calibration, leading to degraded predictive performance or unstable decision boundaries.
- Although the use of the concept is motivated by the desire to provide interpretable model updates, the interpretability of the learned concepts in COMMOD remains questionable. In the current setup, concepts are formed as sparse linear combinations of input features with constraints on diversity and sparsity. However, there is no guarantee that these learned combinations align with human-understandable semantics or domain-relevant groupings. As a result, even if a concept contributes significantly to a model update, it may be difficult for a practitioner or domain expert to interpret the meaning or implication of the change, especially in high-dimensional or non-tabular domains.
- The theoretical results presented in Section 4 provide formal grounding for the feasibility of minimal and fair model updates under certain assumptions, including the derivation of a fairness-aware Bayes-optimal classifier. However, these results remain largely detached from the actual algorithmic design of COMMOD. The analysis presumes access to full knowledge of joint distributions (e.g., over features, labels, and sensitive attributes), which is impractical in real-world applications. Moreover, the Bayes-optimal formulation does not inform the specific architectural choices, training dynamics, or constraint integration in COMMOD. Similarly, the theoretical bounds on fairness change (P-rule improvement) offer little actionable guidance on hyperparameter tuning or trade-off navigation. In essence, while the theory helps justify the problem's tractability, it does not substantially aid or guide the design decisions behind the proposed method, nor does it provide predictive or diagnostic tools to practitioners applying COMMOD.

---

> ### Author Response · Authors · 2025-04-30
> **Answer to reviewer kgV6**
>
> Thank you for your comments and suggestions. We provide below some detailed answers:
>
> # On the potential of COMMOD to handle successive updates over time
> One of the assumptions of COMMOD is that the model $f$ is to be trusted (e.g. by experts validating the model), justifying why enforcing a similarity with $f$ is desirable. In that context, in the face of data drift or evolving objectives, the first question that is to be raised is the extent to which the input model remains trustworthy.
>
> We consider a setting where a first fair model $g$ was trained using COMMOD and an initial model $f$; and a second model $g'$ was to be trained.
>
> - If the predictions of $g'$ cannot be trusted anymore, e.g. in the face of a strong data drift, this goes beyond the scope of the contributions of this paper, which proposes to leverage an accurate (but biased) model to train a fair one. There is always the possibility of possibility of applying COMMOD to an older iteration ($f$) that would be deemed more trustworthy, supposing these iterations would be saved.
> - Similarly, evolving fairness constraints may indeed create a situation where learning $g'$ from $g$ could be problematic; once again, if possible, training on $f$ may be the most suitable solution.
>
>
> # On the human-understandability of COMMOD's concepts
> COMMOD proposes to learn truly “interpretable” concepts in an unsupervised way. It represents each concept as a sparse, linear combination of input features, that can therefore easily inspected by domain experts.
> Although there is always a risk that the learned concepts do not perfectly align with human knowledge:
>
> 1. This limitation is not specific to COMMOD, but inherent to all "unsupervised" interpretability method. Notable examples in the prior literature include: [1, 3]
>
> 2. COMMOD proposes several characteristics known to help with interpretability:
> - Numerous works have proposed to discover such concepts in an unspervised fashion, including in high-dimensional, non-tabular, settings, e.g. images [1,2]
> - Concept discovery for tabular data has also been explored, concurrently to our work [3, 4]
> - Sparse linear models are generally considered to be human-understandable
> - COMMOD's concepts combine sparse representations and diversity, two characteristics that are often seen as favoring human interpretability (and generally considered in concept-based architectures, e.g. [2,3,4])
>
> To conclude, the intuitive definition of COMMOD, defining concepts as sparse linear combinations of features, supports the human-understandability of the learned concepts. We have added some comments and references in Section 5.2 to better motivate our approach.
>
> [1] Ghorbani, A., Wexler, J., Zou, J. Y., & Kim, B. (2019). Towards automatic concept-based explanations. Advances in neural information processing systems, 32.
>
> [2] Yuksekgonul, M., Wang, M., & Zou, J. Post-hoc Concept Bottleneck Models. In The Eleventh International Conference on Learning Representations.
>
> [3] Zarlenga, M. E., Shams, Z., Nelson, M. E., Kim, B., & Jamnik, M. TabCBM: Concept-based Interpretable Neural Networks for Tabular Data. Transactions on Machine Learning Research.
>
> [4] Khurana, U., & Galhotra, S. (2021, October). Semantic concept annotation for tabular data. In Proceedings of the 30th ACM International Conference on Information & Knowledge Management (pp. 844-853).
>
>
> # On the theoretical results
> We agree that these theoretical results requires having knowledge of the true joint distribution or either of the sensitive attributes.
> Yet, they provide some intuitions and understanding about the considered problem:
>
> - Result 1: Although the Bayes-Optimal Classifier (BOC) in Proposition 4.2 assumes full access to the joint distribution \(\mathbb{P}(X,S,Y,\hat{Y})\), it remains invaluable for two reasons. First, it provides a theoretical benchmark: no algorithm can outperform its accuracy–fairness trade-off, so it defines the “limit of the possible” against which all heuristics (including COMMOD) can be measured. Second, the BOC’s analytic form,
> $$
> g_{opt}(x)
> =\;H\Bigl(\eta(x)-c
> \;-\;\lambda_{\rm fair}\bigl[\bar\eta(x)-\bar c\bigr]
> \;-\;\lambda_{\rm ratio}\bigl[\eta^*(x)-c^*\bigr]\Bigr),
> $$
> reveals exactly which terms (accuracy, fairness penalty, change-cost) must be learned and combined, guiding the design of practical surrogates (e.g.adversarial estimators for $\(\bar\eta\)$, $\(\ell_2\)$ penalties on change-rates) that replace idealized distributional quantities with empirical estimates.
>
> - Result 2 also provides some theoretical understanding of the problem, revealing the concrete impact on the fairness criteria of a given number of label flip. It therefore provides an upper bound for the maximum fairness that is reachable in a given setting, which is very valuable when working on a problem, even if our proposed algorithm is not directly derived from it.

---

### Author Response · Authors · 2025-04-30
**Answer to reviewers**

We thank the reviewers for their valuable comments and questions. Besides our answer to each review, we have made several modifications to the paper, which we believe help improving its overall quality. The modifications have been directly integrated to the uploaded version of the article (text in red).

---

### Decision · Action_Editor_bND5 · 2025-06-03

**Recommendation:** Accept with minor revision

**Additional Comments:**

I would like to thank the authors for addressing most of the reviewers' concerns, and I would like these changes to be included in the final version. I would also like to request a few changes before this work gets accepted:

(1) The experimental designs lack statistical significance: they should be repeated with multiple seeds and include confidence intervals, or the reason for avoiding this approach should be mentioned.

(2) The choice of a discrete metric like accuracy when modifying probabilities and its implications should be highlighted more clearly in the empirical section.

**Audience:**

Yes

**Audience Explanation:**

Yes, the paper addresses an interesting question, and the approach will be of interest to the fairness and accountability community. Even the ad hoc theory results can be of independent interest.

**Claims And Evidence:**

Yes

**Claims Explanation:**

The authors claim to provide a framework for debiasing machine learning models in a way that "the differences between new fair model and the existing one should be (i)interpretable and (ii) minimal". The authors propose an algorithm called COMMOD and use this algorithm to empirically validate the claim.

---

> ### Author Response · Authors · 2025-06-25
> **Changes made in the camera-ready version**
>
> Thank you for this decision, and thank you again to the reviewers for their valuable feedback, which helped us make our contribution more solid overall.
>
> Per your request, we have conducted the following changes:
>
> 1) We have added an experiment assessing the sensitivity of COMMOD's results to the random seed. The results, described in Section 6.2 "Sensitivity Analysis", show that the results (accuracy, p_rule, and number of prediction changes) of the final model trained with COMMOD almost exclusively depend on the value of the hyperparameters and the performances of the initial, biased model (used as input). Therefore, resulting sources of randomness (e.g. stochasticity of the neural network initialization) are shown to have a very limited impact.
>
> 2) We have added additional comments in the empirical section (in particular in Section 6.2 "Experiment 1" to justify our choice of using a discrete metrics. Besides, we believe that other changes previously made following the reviewers' comments go into the same direction (e.g. clarification of the optimization objective in Section 3.2.
>
> 3) We have readproofed the paper and performed minor changes (e.g. typos and spelling)